# *Brucella* activates the host RIDD pathway to subvert BLOS1-directed immune defense

Kelsey Michelle Wells[1], Kai He[2], Aseem Pandey[1,3], Ana Cabello[1,3], Dongmei Zhang[1], Jing Yang[1], Gabriel Gomez[4], Yue Liu[5], Haowu Chang[6], Xueqiang Li[6], Hao Zhang[6], Xuehuang Feng[1], Luciana Fachini da Costa[3], Richard Metz[7], Charles D Johnson[7], Cameron Lee Martin[8], Jill Skrobarczyk[8], Luc R Berghman[8], Kristin L Patrick[1], Julian Leibowitz[1], Allison Ficht[9], Sing-Hoi Sze[10,11], Jianxun Song[1], Xiaoning Qian[2,12], Qing-Ming Qin[1,5]\*, Thomas A Ficht[3]\*, Paul de Figueiredo[1,3]\*

[1]Department of Microbial Pathogenesis and Immunology, College of Medicine, Texas A&M Health Science Center, Bryan, United States; [2]Department of Electrical and Computer Engineering, Texas A&M University, College Station, United States; [3]Department of Veterinary Pathobiology, Texas A&M University, College Station, United States; [4]Texas A&M Veterinary Medical Diagnostic Laboratory, Texas A&M University, College Station, United States; [5]College of Plant Sciences, Key Laboratory of Zoonosis Research, Ministry of Education, Jilin University, Jilin, China; [6]Key Laboratory of Symbolic Computation and Knowledge Engineering, Ministry of Education, College of Computer Science and Technology, Jilin University, Changchun, China; [7]Genomics and Bioinformatics Services, Texas A&M University, College Station, United States; [8]Department of Poultry Science, Texas A&M University, College Station, United States; [9]Department of Molecular and Cellular Medicine, College of Medicine, Texas A&M Health Science Center, College Station, United States; [10]Department of Computer Science and Engineering, Dwight Look College of Engineering, Texas A&M University, College Station, United States; [11]Department of Biochemistry & Biophysics, Texas A&M University, College Station, United States; [12]TEES-AgriLife Center for Bioinformatics & Genomic Systems Engineering, Texas A&M University, College Station, United States

**\*For correspondence:**
qmqin@jlu.edu.cn (Q-MQ);
tficht@tamu.edu (TAF);
pjdefigueiredo@tamu.edu (PdF)

**Competing interest:** The authors declare that no competing interests exist.

**Abstract** The phagocytosis and destruction of pathogens in lysosomes constitute central elements of innate immune defense. Here, we show that *Brucella*, the causative agent of brucellosis, the most prevalent bacterial zoonosis globally, subverts this immune defense pathway by activating regulated IRE1α-dependent decay (RIDD) of *Bloc1s1* mRNA encoding BLOS1, a protein that promotes endosome–lysosome fusion. RIDD-deficient cells and mice harboring a RIDD-incompetent variant of IRE1α were resistant to infection. Inactivation of the *Bloc1s1* gene impaired the ability to assemble BLOC-1-related complex (BORC), resulting in differential recruitment of BORC-related lysosome trafficking components, perinuclear trafficking of *Brucella*-containing vacuoles (BCVs), and enhanced susceptibility to infection. The RIDD-resistant *Bloc1s1* variant maintains the integrity of BORC and a higher-level association of BORC-related components that promote centrifugal lysosome trafficking, resulting in enhanced BCV peripheral trafficking and lysosomal destruction, and resistance to infection. These findings demonstrate that host RIDD activity on BLOS1 regulates *Brucella* intracellular parasitism by disrupting BORC-directed lysosomal trafficking. Notably, coronavirus murine hepatitis virus also subverted the RIDD–BLOS1 axis to promote intracellular replication.

Our work establishes BLOS1 as a novel immune defense factor whose activity is hijacked by diverse pathogens.

## Editor's evaluation

To successfully replicate in the host cell, Brucella must evade degradation in lysosomes and traffic to the ER. This work uncovers a novel mechanism by which Brucella harnesses the host unfolded protein response to degrade Blos 1, a key regulator of lysosomal trafficking, thereby enabling pathogenic Brucella peri-nuclear/ER trafficking.

## Introduction

*Brucella* is an intracellular vacuolar pathogen that invades many cell and tissue types, including nonprofessional and professional phagocytes (*de Figueiredo et al., 2015*). Brucellosis has eluded systematic attempts at eradication for more than a century (*Godfroid et al., 2002*), and even in most developed countries, no approved human vaccine is available (*Ficht and Adams, 2009*). The intracellular lifestyle limits exposure to host innate and adaptive immune responses and sequesters the organism from the effects of some antibiotics. *Brucella* evades intracellular destruction by limiting interactions of the *Brucella*-containing vacuole (BCV) with the lysosomal compartment (*Criscitiello et al., 2013*; *Pizarro-Cerdá et al., 1998*). BCVs harboring internalized *Brucella* traffic from endocytic compartments (eBCVs) to a replicative niche within vacuoles (rBCVs) that are decorated with markers of the endoplasmic reticulum (ER) (*Pizarro-Cerdá et al., 1998*; *Starr et al., 2012*). BCVs also accumulate autophagic membranes (aBCVs), which constitute a distinctive aspect of the intracellular lifestyle of the pathogen (*Pandey et al., 2018*; *Starr et al., 2012*). The VirB type IV secretion system (T4SS) is a significant virulence factor that regulates *Brucella* intracellular trafficking (*Marchesini et al., 2011*; *Paredes-Cervantes et al., 2011*; *Sá et al., 2012*; *Smith et al., 2012*). *Brucella* effectors secreted by the T4SS promote bacterial intracellular trafficking and growth via modulation of host functions (*de Barsy et al., 2011*; *de Jong et al., 2008*; *Döhmer et al., 2014*; *Miller et al., 2017*; *Myeni et al., 2013*) and organisms that lack this system fail to establish productive infections.

The Unfolded Protein Response (UPR) is an evolutionarily conserved signaling pathway that allows the ER to recover from the accumulation of misfolded proteins (*Gardner et al., 2013*; *Walter and Ron, 2011*) during ER stress. The UPR signals through the stress sensors IRE1α, ATF6, and PERK located in the ER membrane. When the luminal domains of these proteins sense unfolded proteins, they transduce signals to their cytoplasmic domains, which initiate signaling that ultimately results in UPR (*Lee et al., 2008*). IRE1α plays a central role in triggering UPR through an endonuclease/RNase activity in its cytoplasmic tail that catalyzes the splicing of *Xbp1* mRNA, which is then translated to generate the XBP1 transcription factor (*Lee et al., 2008*; *Ron and Walter, 2007*). IRE1α RNase activity can also cleave a wide variety of cellular mRNAs that leads to their degradation in a process termed regulated IRE1-dependent mRNA decay (RIDD) (*Hollien and Weissman, 2006*). The RIDD pathway displays selectivity. For example, the pathway cleaves a specific subset of mRNAs encoding polypeptides destined for cotranslational translocation into the ER lumen. The degradation of these mRNAs supports ER homeostasis by reducing the flux of nonessential polypeptides into the ER (*Hollien and Weissman, 2006*). The molecular targets of RIDD activity, and the physiological roles that this process plays in cells, remain areas of investigation.

*Brucella* infection induces host cell ER stress and activates host UPR (*de Jong et al., 2013*; *Pandey et al., 2018*; *Smith et al., 2013*; *Taguchi et al., 2015*; *Wang et al., 2016*). The UPR sensor IRE1α, but neither PERK nor ATF6, is required for the intracellular replication of the pathogen (*Qin et al., 2008*; *Taguchi et al., 2015*), indicating that the IRE1α signaling pathway confers susceptibility to host cell parasitism. An IRE1α–ULK1 signaling axis also contributes to conferring susceptibility to *Brucella* intracellular replication; IRE1α-directed activation of components of the host autophagy program promotes proper bacterial intracellular trafficking and replication (*Pandey et al., 2018*). Despite the abovementioned advances, our understanding of how the IRE1α–RIDD axis and downstream processes regulate the intracellular lifestyle of *Brucella* remains largely unknown.

BLOS1 [biogenesis of lysosome-related organelles complex-1 (BLOC-1) subunit 1, also known as BLOC1S1/GCN5L1], a subunit of both the BLOC-1 and the BLOC-1-related complex (BORC), plays

diverse roles in cells, including mitochondrial protein acetylation, modulation of metabolic pathways, and endosome–lysosome trafficking and fusion (*Bae et al., 2019*; *Guardia et al., 2016*; *Pu et al., 2017*; *Pu et al., 2015*). In mammalian cells, BLOS1 has also been shown to be a principal target of RIDD activity (*Bae et al., 2019*; *Bright et al., 2015*; *Hollien et al., 2009*) and is required for host cell cytotoxicity induced by Ebola virus (*Carette et al., 2011*). To date, the roles and mechanisms by which BLOS1 controls infection by intracellular pathogens remain largely unknown. Here, we demonstrate that IRE1α-directed *Bloc1s1* mRNA degradation confers susceptibility to *Brucella* infection. *Brucella*-induced RIDD activity suppresses *Bloc1s1* expression, disassembles BORC components, and limits BLOS1-regulated interactions between BCVs and lysosomes. In addition, we show that murine hepatitis virus (MHV), a betacoronavirus, also subverts BLOS1 activity to promote its intracellular replication. Collectively, these activities promote the productive subcellular trafficking and intracellular replication of diverse pathogens. Our findings, therefore, identify BLOS1 as a novel immune defense factor that defends against bacterial and viral infection and show that *Brucella* and MHV subvert this innate immune defense system to promote disease.

## Results

### *Ern1* conditional knockout mice are resistant to *Brucella* infection

*Brucella* induces host cell UPR during infection (*Qin et al., 2008*; *Taguchi et al., 2015*) and activates an IRE1α-to-autophagy signaling axis in host cells to promote its intracellular lifestyle (*Pandey et al., 2018*). To extend these findings to an in vivo model of brucellosis, we tested the hypothesis that UPR and IRE1α confer susceptibility to *Brucella* infection in mice harboring a conditional mutation in *Ern1*, the gene encoding IRE1α. Because mice homozygous for null mutations in *Ern1* display embryonic lethality during organogenesis, we used a control *Lyz2-Ern1*^wt/wt [henceforth, wt (wild-type)-IRE1α] and a *Ern1* conditional knockout (CKO mouse line (*Ern1*^flox/flox; *Lyz2*^Cre/+, hence after, m-IRE1α)) (*Figure 1—figure supplement 1A*) (gift from the Iwawaki lab) in the experiments. In this line, exons 20–21 of the gene encoding IRE1α were deleted in monocytes and macrophages, generating animals in which the endonuclease domain (and hence RIDD activity) was specifically disrupted. However, the kinase domain remained intact (*Hur et al., 2012*; *Iwawaki et al., 2009*). Macrophages are critical cellular targets for *Brucella* colonization (*de Figueiredo et al., 2015*). Hence, the tissue and molecular specificity of this lesion rendered the m-IRE1α mouse an ideal system for investigating how bacterial activation of host RIDD activity controls intracellular parasitism by the virulent *B. melitensis* strain 16M (Bm16M).

After confirming the RNase activity deficiency of the truncated IRE1α in bone marrow-derived macrophages (BMDMs) from m-IRE1α animals (*Figure 1—figure supplement 1B*), we verified that m-IRE1α mice had normal organ morphologies, fertility, growth, and development. In addition, we showed that these animals had similar B cell (B220^+), T cell (CD4^+ or CD8^+), and CD11b^+ profiles (*Figure 1—figure supplement 1C*). We also showed that the IL-1β, IL-6, and TNF-α responses of BMDMs to LPS stimulation were reduced in mutant mice (*Figure 1A–C*), consistent with previous findings that these LPS-mediated responses are controlled, in part, by IRE1α activity (*Martinon et al., 2010*).

To determine whether IRE1α activity in macrophages contributed to pathogen burden, dissemination, and disease progression, we infected wt-IRE1α control and m-IRE1α mice with Bm16M via the intraperitoneal route, humanely sacrificed the mice at various times postinfection, and then determined the bacterial burden in assorted tissues by quantifying the number of recovered colony-forming units (CFUs). We found that tissue-specific mutation of IRE1α resulted in enhanced resistance to bacterial infection with significant reductions in bacterial load in the spleen and liver compared to wt-IRE1α controls at 7 or 14 days postinfection (dpi), respectively (*Figure 1D, E*). However, both infected wt- and m-IRE1α mice displayed similar spleen weights (*Figure 1—figure supplement 1D, E*) and spleen or liver inflammation (*Figure 1F–H*), revealing that the lower numbers of CFU recovered from m-IRE1α animals were not accompanied by corresponding decreases in inflammation. To test the hypothesis that the differential bacterial burden in macrophage cells, the predominant cell type in which the pathogen resides and replicates in vivo, accounted for this reduction, we compared the bacterial load in CD11b^+ cells from control and m-IRE1α mice that had been infected with Bm16M for 7 days. We found that indeed CD11b^+ cells from the spleens of m-IRE1α mice displayed striking

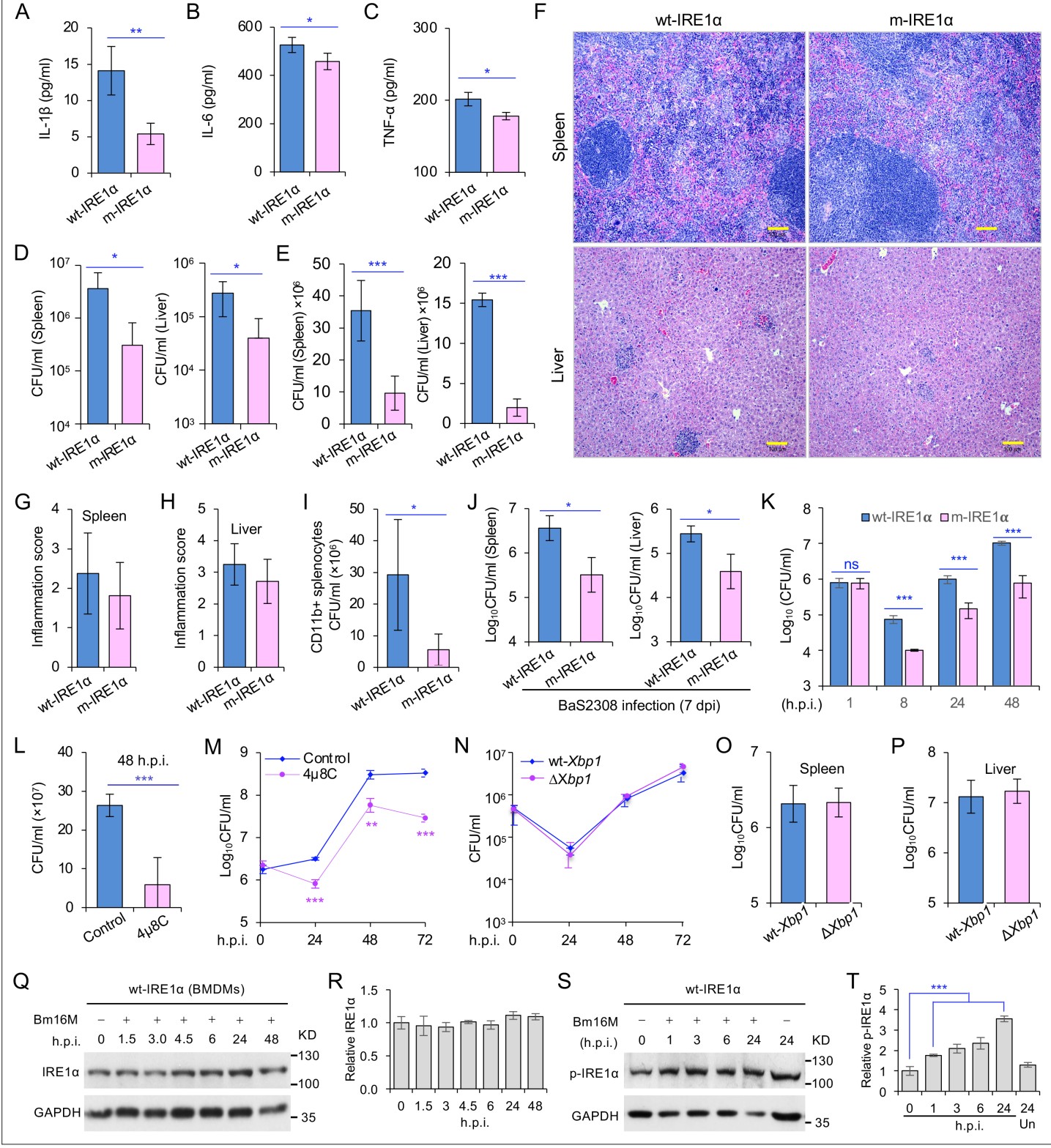

**Figure 1.** Host IRE1α is required for *Brucella* infection in vivo. Innate cytokine production of IL-1β (**A**), IL-6 (**B**), and TNF-α (**C**) in bone marrow-derived macrophages (BMDMs) from the wild-type (WT, wt-IRE1α) control and *Ern1* conditional knockout (CKO, m-IRE1α) mice. The BMDMs were stimulated with LPS (100 ng/ml) and at 6 hr poststimulation the cytokine production of the treated cells was determined. Colony-forming unit (CFU) assay for *B. melitensis* 16M (Bm16M) intracellular survival in spleens and livers of wt- and m-IRE1α mice at 7 (**D**) or 14 (**E**) days post infection (dpi). (**F**) Histopathology of representative hematoxylin and eosin (H&E) stained sections of spleen and liver from Bm16M-infected wt- and m-IRE1α mice at 14 dpi. Bar: 100 μm.

*Figure 1 continued on next page*

*Figure 1 continued*

Quantification of inflammation of spleens (**G**) or livers (**H**) at 14 dpi. (**I**) CFU assays of CD11b$^+$ cells from Bm16M-infected wt- or m-IRE1α mice. (**J**) CFU assay for *B. abortus* S2308 (BaS2308) intracellular survival in spleens and livers in wt-IRE1α control or m-IRE1α mice at 7 dpi. (**K**) Bm16M invasion and intracellular replication in BMDMs from m-IRE1α and control mice. h.p.i.: hours post infection. CFU assays of Bm16M infection of WT BMDMs (**L**) or RAW264.7 macrophages (**M**). Host cells were pretreated with 4µ8C (50 µM) 1 hr before and during infection; CFUs of the infected cells were determined at the indicated h.p.i. (**N**) CFU assays for Bm16M infection of BMDMs from WT and *Xbp1* knockout (Δ*Xbp1*) mice at the indicated h.p.i. CFU assay for Bm16M intracellular survival in spleen (**O**) or liver (**P**) in WT or Δ*Xbp1* mice at 14 dpi. Immunoblotting assay for IRE1α expression (**Q**) and quantification of the expression levels (**R**) in BMDMs during a time course (48 hr) of Bm16M infection. Bm16M infection induces phosphorylation of host IRE1α (**S**) and quantification of the phosphorylated levels of IRE1α during a time course (24 hr) of infection (**T**). Images/blots are representative of three independent experiments. Statistical data represent the mean ± standard error of mean (SEM) from three independent experiments. *, **, and *** indicate significance at p < 0.05, 0.01, and 0.001, respectively.

The online version of this article includes the following figure supplement(s) for figure 1:

**Figure supplement 1.** Characterization of *Ern1* conditional knockout (CKO) and control mice.

**Figure supplement 2.** IRE1α is required for *B. melitensis* intracellular replication.

reductions in bacterial load (*Figure 1I*), thereby suggesting that the resistance of these cells to intracellular parasitism contributed to the resistance phenotype observed at the organismal level. Divergent *Brucella* species display distinct host preferences; however, their interactions with host cells share common features (*de Figueiredo et al., 2015*). To test the hypothesis that m-IRE1α mice also displayed resistance to infection by other *Brucella* species, we infected these mice with *B. abortus* strain S2308 (BaS2308), a strain that displays tropism for cattle. We then assessed tissue burden in spleen and liver at 7 dpi. We found that m-IRE1α mice also exhibited resistance to BaS2308 infection (*Figure 1J*), thereby indicating that the resistance phenotype of the mutant mice was not pathogen species specific.

## IRE1α RNase activity confers susceptibility to *Brucella* infection

*Xbp1* splicing was dramatically diminished in BMDMs from m-IRE1α mice (*Figure 1—figure supplement 1B*), indicating that BMDMs from the m-IRE1α mice carried the expected functional defects in IRE1α RNase activity. We thus tested the hypothesis that IRE1α RNase activity confers susceptibility to intracellular parasitism by Bm16M. First, we performed CFU assays of Bm16M infection of BMDMs from m-IRE1α and control mice and found that the replication efficiency of Bm16M in m-IRE1α BMDMs at 8, 24, and 48 h.p.i. was significantly lower than controls (*Figure 1K*). Similar results were observed in control and *Ern1*$^{-/-}$ mouse embryonic fibroblasts (MEFs) (*Qin et al., 2008*; *Figure 1—figure supplement 2A-B*). Second, we observed fewer Bm16M were recovered from BMDMs or RAW264.7 macrophages treated with 4µ8C, a compound that specifically antagonizes IRE1α RNase activity while leaving its kinase activity intact (*Cross et al., 2012*), than mock-treated controls (*Figure 1L–M*). These findings supported the hypothesis that IRE1α RNase activity confers susceptibility to infection by regulating the intracellular trafficking and/or survival of the pathogen.

## IRE1α regulation of Bm16M intracellular replication is XBP1 independent

We performed additional experiments to interrogate the role of IRE1α RNase activity in controlling *Brucella* infection. Two possibilities were explored: (1) IRE1α catalyzed splicing of *Xbp1* transcripts, and downstream expression of XBP1-responsive genes, conferred susceptibility to Bm16M infection; or (2) IRE1α catalyzed RIDD activity controlled this process. To determine whether *Xbp1* splicing was the reason, we examined the survival and intracellular replication of the pathogen in BMDMs harvested from *Xbp1*$^{-/-}$ mice (gift from the Glimcher lab) in which *Xbp1* was conditionally ablated from monocytes, macrophages, and granulocytes (henceforth, Δ*Xbp1* mice) and found that Bm16M replicated similarly in Δ*Xbp1* BMDMs and wt-*Xbp1* littermate controls (*Figure 1N*). Moreover, Bm16M displayed similar levels of liver and spleen colonization in Δ*Xbp1* and wt-*Xbp1* mice (*Figure 1O, P*). These data suggested an XBP1-independent role for IRE1α RNase activity in conferring susceptibility to Bm16M infection, thereby implicating RIDD as the sought-after activity (see below).

## IRE1α activity regulates *Brucella* intracellular trafficking and replication

To dissect the mechanism by which IRE1α activity confers susceptibility to intracellular parasitism by Bm16M, we examined this process in BMDMs derived from m-IRE1α mice. We observed that the expression level of IRE1α was relatively unchanged during a time course (48 hr) of infection in wt- or m-IRE1α BMDMs and MEFs (*Figure 1Q, R*; *Figure 1—figure supplement 2C-D*). As expected, CKO of IRE1α did not impair IRE1α phosphorylation in response to *Brucella* infection (*Figure 1— figure supplement 2E*). IRE1α phosphorylation was enhanced over the same time course in wt-IRE1α BMDMs during Bm16M infection (*Figure 1S,T*). Host IRE1α activity was also required for *Brucella* intracellular parasitism (*Figure 1K*; *Figure 1—figure supplement 2*). To test whether IRE1α regulates Bm16M intracellular trafficking, we used confocal immunofluorescence microscopy (CIM) to analyze the localization of the pathogen in *Ern1*$^{+/+}$ and *Ern1*$^{-/-}$ MEFs, or m-IRE1α and control BMDMs. In IRE1α harboring controls, the pathogen transiently trafficked through early and late endosomes (EEA1$^+$ and M6PR$^+$ compartments, respectively) (*Figure 2A–D*; *Figure 2—figure supplement 1A–D*) before primarily accumulating (at 24 and 48 h.p.i.) in a replicative niche decorated with the ER marker calreticulin (*Figure 2E, F*; *Figure 2—figure supplement 1E and G*); however, in m-IRE1α BMDMs or *Ern1*$^{-/-}$ MEF cells, Bm16M displayed reduced trafficking to calreticulin$^+$ compartments (*Figure 2E, F*; *Figure 2—figure supplement 1E, G*). Instead, the pathogen trafficked with greater efficiency to M6PR$^+$ late endosomes (at 12 h.p.i.) (*Figure 2B, D*; *Figure 2—figure supplement 1B, D*), and to LAMP1$^+$ or cathepsin D$^+$ lysosomes (at 24 and 48 h.p.i.) (*Figure 2G–J*; *Figure 2—figure supplement 1F, H*). Our data, therefore, demonstrated that IREα activity controls Bm16M intracellular replication, likely via regulation of BCV ER trafficking. These findings encouraged us to investigate the molecular mechanisms driving these phenomena.

### *Brucella* infection downregulates RIDD genes

We were intrigued with the hypothesis that Bm16M subverts host RIDD activity to promote intracellular parasitism. First, since host UPR/IRE1α RNase activity is induced by *Brucella* effectors secreted by the T4SS of the pathogen (*de Jong et al., 2013*), we tested whether RIDD activity was dependent upon the *Brucella* T4SS. We found that induction of IRE1α RNase activity occurred in a *Brucella* T4SS-dependent fashion (*Figure 3—figure supplement 1A*). Next, we performed RNA-seq analysis to define candidate host genes whose transcripts were subject to RIDD control during *Brucella* infection. Specifically, we used Bm16M to infect triplicate sets of BMDMs as follows: (1) solvent control-treated wt-IRE1α, (2) 4μ8C-treated wt-IRE1α, or (3) solvent control-treated m-IRE1α (*Figure 3—figure supplement 1B*). At 4 or 24 h.p.i., we harvested host mRNA for RNA-seq analysis. Differential expression analysis was then performed to identify genes that were downregulated following Bm16M infection of wt-IRE1α cells but were unchanged or upregulated in either infected, drug-treated cells, or infected, m-IRE1α cells. Genes that displayed reduced expression ($p < 0.05$) in response to infection at 4 and/or 24 h.p.i., and also whose infection-dependent reductions in expression were reversed upon treatment with 4μ8C, or in m-IRE1α cells, were defined as candidate 'RIDD genes'. This analysis resolved 847 candidate RIDD genes (*Figure 3A–C*; *Figure 3—figure supplement 1C–F*; *Figure 3—figure supplement 2*; *Figure 3—source data 1*). KEGG pathway and interaction network analyses revealed that most RIDD candidate genes were involved in cellular component organization and biogenesis, RNA metabolism, and oxidative phosphorylation (*Figure 3C*; *Figure 3—figure supplement 2*).

We performed several experiments or analyses to validate candidate RIDD genes identified in the RNA-seq analysis. First, we compared our list of candidate RIDD genes to genes previously reported to be subject to RIDD control (*Bright et al., 2015*; *Han et al., 2009*; *Hollien et al., 2009*; *So et al., 2012*). This comparison identified 40 genes that were previously shown to be substrates of IRE1α RNase activity and/or displayed expression patterns consistent with RIDD targeting (*Figure 3A, B*). Second, we used real-time quantitative reverse transcription-PCR (qRT-PCR) to measure the expression levels of several candidate RIDD genes, including *Bloc1s1*, *Cd300lf*, *Diras2*, and *Txnip* (*Figure 3D–G*; *Figure 3—figure supplement 3*) during infection. We found that the expression of these genes was significantly lower in *Brucella*-infected wt-IRE1α cells than in m-IRE1α cells (*Figure 3D–G*). Third, we examined whether similar reductions in expression of candidate RIDD genes were observed in host cells infected with *B. abortus* S19 (BaS19, a vaccine strain) or BaS2308. We found that BaS19 or BaS2308 induced similar phenotypes as Bm16M (*Figure 3—figure supplement 3A–D*), suggesting that the phenotype was not species specific. To determine whether RIDD-mediated Txnip expression

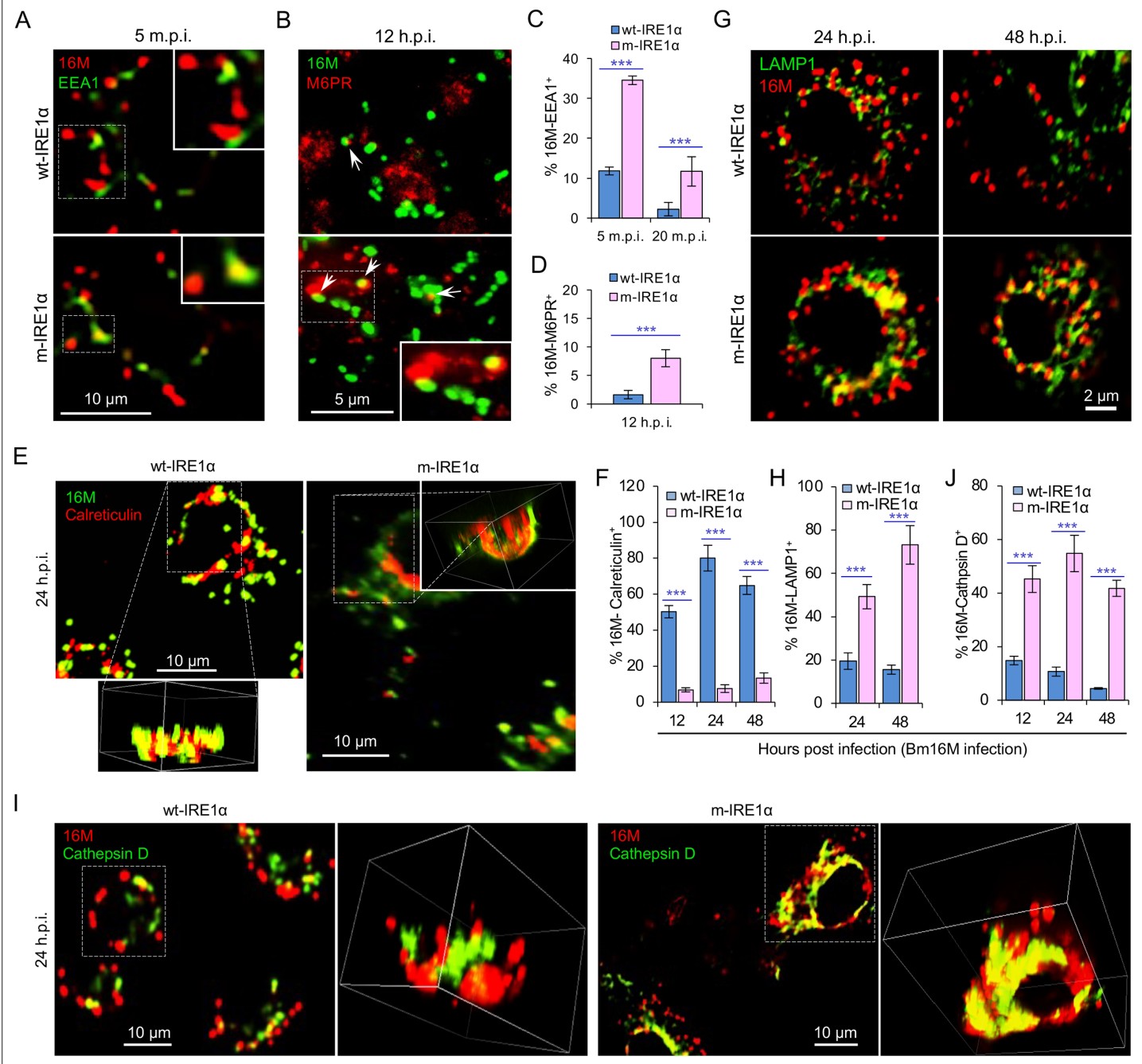

**Figure 2.** IRE1α regulates proper intracellular trafficking and replication of *Brucella* in a XBP1-independent fashion. Colocalization analysis of Bm16M with host early endosomes (**A**) and late endosomes (**B**) of BMDMs from wt- and m-IRE1α mice at the indicated time points postinfection. m.p.i.: minutes post infection. EEA1: early endosomal antigen 1; M6PR: mannose-6-phosphate receptor. Arrows in panel B: M6PR⁺-BCVs. Quantification of Bm16M entry into early endosomes (**C**) or late endosomes (**D**) of the indicated host BMDMs at the indicated time points postinfection. Colocalization analysis of Bm16M and the ER marker calreticulin (**E**), and quantification of Bm16M-calreticulin⁺ (**F**) in wt- and m-IRE1α BMDMs at the indicated h.p.i. Colocalization of Bm16M and the lysosomal markers LAMP1 (**G**) or cathepsin D (**I**), and quantification of Bm16M-LAMP1⁺ (**H**) or -cathepsin D⁺ (**J**) in wt- and m-IRE1α BMDMs at the indicated h.p.i. Images are representative of three independent experiments. Statistical data express as mean ± standard error of mean (SEM) from three independent experiments. ***p < 0.001.

The online version of this article includes the following figure supplement(s) for figure 2:

**Figure supplement 1.** IRE1α is required for *B. melitensis* properly intracellular trafficking.

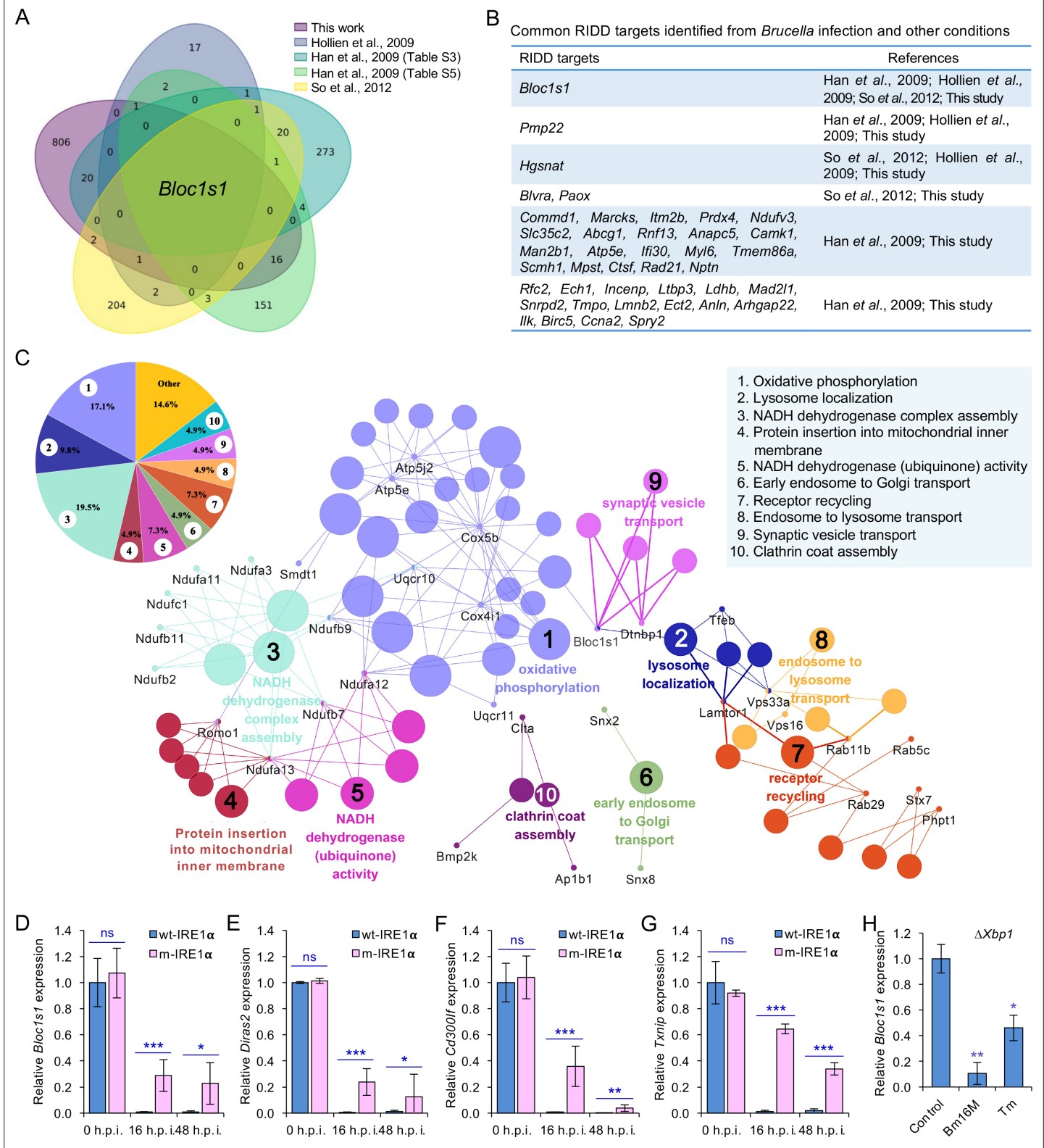

**Figure 3.** Identification of host RIDD targets during *Brucella* infection. (**A**) Venn diagram showing numbers of candidate RIDD genes identified in the indicated datasets. (**B**) Common candidate RIDD genes identified in Bm16M-infected cells and other conditions in the indicated datasets. (**C**) Interaction network analysis of candidate RIDD genes (*Bloc1s1* associated genes) identified in *Brucella*-infected cells and the corresponding enriched KEGG pathways. Different pathways are distinguished by different colors. Interacting genes are shown with the smallest sized of dot with gene names. The

*Figure 3 continued on next page*

*Figure 3 continued*

upper-left-corner panel: enriched KEGG pathways and interacting candidate RIDD genes (%). qRT-PCR validation of RIDD candidate genes *Bloc1s1* (**D**), *Diras2* (**E**), *Cd300lf* (**F**), and *Txnip* (**G**) identified from RNA-seq analysis. Relative mRNA expression levels in potential RIDD targets from control and m-IRE1α BMDMs infected with Bm16M at 16 and 48 h.p.i. were measured by qRT-PCR. (**H**) qRT-PCR analysis of expression levels of *Bloc1s1* in Δ*Xbp1* BMDMs that were either uninfected (control), infected with Bm16M, or treated with tunicamycin (Tm, an UPR inducer, 5 µg/ml) at 4-hr post infection/ treatment. Expression levels of the indicated genes were normalized to *Gapdh* expression. Statistical data represent the mean ± standard error of mean (SEM) from three independent experiments. *, **, and ***: significance at p < 0.05, 0.01, and 0.001, respectively.

The online version of this article includes the following source data and figure supplement(s) for figure 3:

**Source data 1.** Candidate RIDD gene identified in host cells infected by Brucella melitensis Bm16M.

**Figure supplement 1.** IRE1α activation is *Brucella* Type 4 secretion system (T4SS)-dependent and gene profiling of host cells infected by *Brucella*.

**Figure supplement 2.** KEGG pathway network analysis of the candidate RIDD genes identified via RNA-seq analysis from host cells infected or uninfected with Bm16M and/or treated or untreated with 4µ8C at 4 and/or 24 h.p.i.

**Figure supplement 3.** Validation of RIDD target genes.

was also regulated by the microRNA miR17, a molecule that affects TXNIP mRNA stability (*Lerner et al., 2012*), we measured the expression of miR-17 via qRT-PCR. We found that the expression of miR-17 was reduced (*Figure 3—figure supplement 3E*). Fourth, we used qRT-PCR to test the hypothesis that heat-killed bacteria induced similar changes in RIDD gene expression. We found that heat-killed bacteria did not cause a similar effect. Hence, the induction of RIDD activity in host cells required interactions with the viable agent (*Figure 3—figure supplement 3F*) and was also T4SS dependent (*Figure 3—figure supplement 1A*). Finally, we found that the expression of the key RIDD gene *Bloc1s1* was also reduced in Δ*Xbp1* BMDMs infected with Bm16M, or when ER stress was induced in these cells (*Figure 3H*). These data suggested that the observed changes in host gene expression patterns were not a consequence of alterations in XBP1 transcription factor activity. Taken together, these data supported the hypotheses that (1) Bm16M infection induces RIDD activity in host cells and (2) RIDD activity confers enhanced susceptibility to intracellular parasitism by *Brucella*. However, these findings left open the question of the molecular mechanism by which RIDD activity controlled Bm16M replication.

## RIDD activity on *Bloc1s1* controls *Brucella* intracellular parasitism

Our RNA-seq analysis identified *Bloc1s1* as a *Brucella*-induced RIDD gene. BLOS1 is a subunit of both the BLOC-1 and BORC complexes and plays diverse roles in cell physiological and biological processes, including endosome–lysosome trafficking and fusion (*Figure 3C*; *Bae et al., 2019*; *Guardia et al., 2016*; *Pu et al., 2017*; *Pu et al., 2015*). However, the mechanisms by which *Bloc1s1* regulates microbial infection are largely unknown. This fact encouraged us to test the hypothesis that *Bloc1s1* plays a central role in regulating Bm16M intracellular parasitism. First, we generated a cell line carrying a nonfunctional *Bloc1s1* mutant allele (m*Bloc1s1*). Mammalian BLOS1 contains three conserved XAT hexapeptide-repeat motifs that are essential for acetyltransferase activity and may also be a necessary structure-defining feature for acetyl-CoA contact (*Scott et al., 2018*; *Wu et al., 2021a*). Using CRISPR/cas9-mediated gene editing, we mutated the first XAT hexapeptide-repeat motif, which in the wild-type encodes 'EALDVH', and in the mutant encodes 'EVVDH or EVDH' (*Figure 4—figure supplement 1A*, *Supplementary file 1*). A cell line containing gene encoding Cas9 and a nonspecific gRNA was used as a control of the m*Bloc1s1* mutant line. Second, we generated a RIDD-resistant *Bloc1s1* cell line (henceforth Rr-*Bloc1s1*). In this line, a mutation (from 'G' to 'U') was introduced into *Bloc1s1* mRNA stem-loop structure (i.e., the target of IRE1α RNase activity) that rendered the mutated *Bloc1s1* mRNA stem-loop structure-resistant cleavage by IRE1α RNase. A *Bloc1s1::Bloc1s1-HA* line that overexpresses *Bloc1s1* (wt-*Bloc1s1*) served as a control of the Rr-*Bloc1s1* cell line (*Figure 4— figure supplement 1B*; *Supplementary file 1*).

We characterized the developed cell lines in several ways. First, we noted that α-tubulin acetylation levels had been reported to be controlled, in part, by BLOS1 activity levels (*Wu et al., 2018*). Therefore, we monitored α-tubulin acetylation to assess whether our developed cell lines did, in fact, display alterations in BLOS1 activity. We found that m*Bloc1s1* and Rr-*Bloc1s1* cells displayed reduced levels (*Figure 4A*) and maintained relatively higher levels (*Figure 4B*), respectively, of acetylated α-tubulin, compared to their corresponding controls. These data supported the hypothesis that these

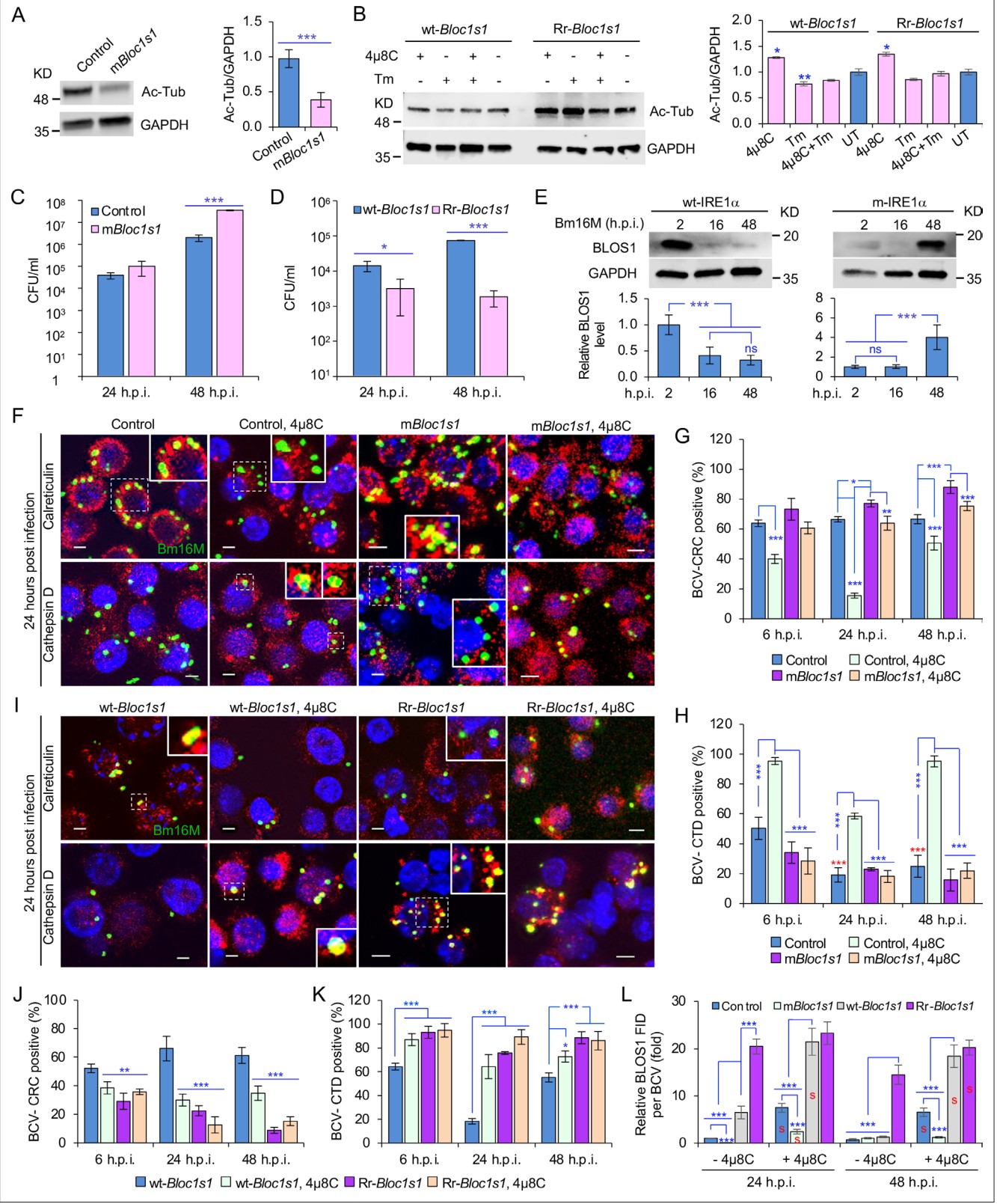

**Figure 4.** BLOS1 confers host cell susceptibility to *Brucella* infection and controls *Brucella* intracellular trafficking. (**A**) Western blot analysis of α-tubulin acetylation (left) and quantification of α-tubulin acetylation level (right) in control containing Cas9 and a nonspecific gRNA and the nonfunctional *Bloc1s1* mutant (m*Bloc1s1*) in RAW 264.7 Cas9 cells. Ac-Tub: anti-acetylated antibody. (**B**) Western blot analysis of α-tubulin acetylation (left) and quantification of α-tubulin acetylation levels (right) in control (wt-*Bloc1s1*, overexpressing WT *Bloc1s1*) cells and cells that express the RIDD-resistant

*Figure 4 continued on next page*

*Figure 4 continued*

*Bloc1s1* variant (Rr-*Bloc1s1*) treated or untreated with 4μ8C (50 μM), Tm (5 μg/ml), or both for 4 hr. CFU assays for Bm16M infection of RAW264.7 cells in which *Bloc1s1* is nonfunctional (**C**), RIDD resistant (**D**) at the indicated h.p.i. (**E**) BLOS1 degradation assay during *Brucella* infection (upper panel) and quantification of the relative BLOS1 expression level (compared to the level of the loading control GAPDH) (lower panel) at the indicated h.p.i. ns: no significance. Colocalization of BCV with calreticulin (CRC) or cathepsin D (CTD) (**F**) and quantification of BCV-CRC⁺ (**G**) or BCV-CTD⁺ (**H**) in control and m*Bloc1s1* cells treated with or without 4μ8C (50 μM) at the indicated h.p.i. Red asterisks: significance (p < 0.001) compared to control cells at 6 h.p.i. Colocalization of BCV with CRC or CTD (**I**) and quantification of BCV-CRC⁺ (**J**) or BCV-CTD⁺ (**K**) in the wt-*Bloc1s1* and Rr-*Bloc1s1* cells treated with or without 4μ8C (50 μM) at the indicated h.p.i. (**L**) Quantification of BLOS1 fluorescence integrated density (FID) per BCV in the m*Bloc1s1*, Rr-*Bloc1s1*, or their corresponding control cells treated with or without 4μ8C (50 μM) at the indicated h.p.i. S: significance (p < 0.01) compared to that without 4μ8C treatment. Host cells were infected with or without Bm16M, and at the indicated h.p.i., the cells were harvested for immunoblotting assays or fixed and subjected to confocal immunofluorescence assays. Blots/images are representative of three independent experiments. Statistical data represent the mean ± standard error of mean (SEM) from three independent experiment. *p < 0.05; **p < 0.01; ***p < 0.001.

The online version of this article includes the following figure supplement(s) for figure 4:

**Figure supplement 1.** Generation of nonfunctional and overexpression *Bloc1s1* variants.

**Figure supplement 2.** Cells with BLOS1 deficiency or RIDD resistance differently control lysosome intracellular trafficking.

**Figure supplement 3.** Host endogenous BLOS1 and the associated proteins are specifically recognized by the indicated homemade or commercial antibodies and differential interactions of *Brucella* and host BLOS1 during infection.

cells had the expected levels of BLOS1 activity. Second, we tested the replication of the pathogen in different *Bloc1s1* cell lines. We found that m*Bloc1s1* cells exhibited increased susceptibility to Bm16M infection (***Figure 4C***), whereas Rr-*Bloc1s1* cells or wild-type controls treated with 4μ8C supported dramatically reduced intracellular bacterial growth (***Figures 1M and 4D***; ***Figure 4—figure supplement 1C***). Finally, we monitored the expression levels of BLOS1 protein during a 48-hr time course of infection. We found that BLOS1 expression was reduced at 16 h.p.i., and continuously decreased during Bm16M infection in wt-IRE1α control cells; however, in m-IRE1α BMDMs, BLOS1 expression was relatively stable or increased (at 48 h.p.i.) (***Figure 4E***). Similar results were observed in 4μ8C treated or untreated m*Bloc1s1*, Rr-*Bloc1s1*, and control cells infected with BaS2308 (***Figure 4—figure supplement 1D, E***). These data demonstrate that low or high BLOS1 expression levels promote or impair *Brucella* infection, respectively.

## BLOS1 regulates *Brucella* intracellular trafficking

The mechanism by which BLOS1 regulates *Brucella* infection was unknown. However, the observed subcellular trafficking defect of the pathogen in host cells harboring mutant or deficient variants of IREα (***Figure 2***; ***Figure 2—figure supplement 1***) suggested that BLOS1 may control the intracellular parasitism of the pathogen by regulating its subcellular trafficking. To illuminate this aspect, we first characterized the m*Bloc1s1*, Rr-*Bloc1s1*, and the corresponding control cell lines by treating them with tunicamycin (Tm, an UPR inducer) or 4μ8C, or infected them with Bm16M. We then assessed the trafficking of the pathogen in these cells using CIM. We found that low levels of BLOS1 or nonfunctional BLOS1 in uninfected or infected cells were associated with the accumulation of late endosome/lysosome (LE/Lys) membranes in the vicinity of nuclei, reduced colocalization of latex beads with cathepsin D, and increased perinuclear LC3b index or autophagic activity near nuclei, in both control and Tm-treated conditions (***Figure 4—figure supplement 2A, C, E, G***). In these studies, the LC3b index was defined as: (Total number of identified cells with the ratio of the mean LC3b intensity in the cytoplasm to that in the nucleus <1)/(Total number of the analyzed cells). In contrast, overexpression of BLOS1 (the wild-type cells expressing wt-*Bloc1s1* or Rr-*Bloc1s1*) reduced LE/Lys perinuclear accumulation, increased the localization of latex beads in cathepsin D⁺ compartments, and reduced perinuclear autophagic activity (***Figure 4—figure supplement 2B, D, F, H***). These data indicated that cells deficient in BLOS1 or with RIDD resistance differentially control lysosome intracellular trafficking and autophagic activity. Although significant inhibition of BCV trafficking to lysosomes in the m*Bloc1s1* cells was not observed at 24 and 48 h.p.i., the m*Bloc1s1* cells supported enhanced BCV trafficking to ER compartments during bacterial infection, compared to that in the wild-type control cells, or 4μ8C-treated m*Bloc1s1* cells (***Figure 4F–H***). In contrast, Rr-*Bloc1s1* cells displayed reduced BCV trafficking to ER compartments, but instead promoted BCVs trafficking to lysosomes during Bm16M infection, compared to controls (***Figure 4I–K***).

To test the hypothesis that Bm16M infection alters the dynamics of associations between BLOS1 and BCVs, we used CIM approaches to localize these elements during a time course of infection after confirmation of the specificities of antibodies used in the work (see below) (*Figure 4—figure supplement 3A–D*). We found higher levels of BLOS1 colocalization with BCVs in 4µ8C-treated control or m*Bloc1s1* cells than their corresponding untreated cells (*Figure 4L*); moreover, at 24 h.p.i., lower levels of BLOS1$^+$ BCVs were observed in m*Bloc1s1* cells compared to wild-type controls (*Figure 4—figure supplement 3E, F*). Rr-*Bloc1s1* and/or 4µ8C inhibition of *Bloc1s1* degradation in host cells (i.e., Rr-*Bloc1s1*, 4µ8C-treated wt-*Bloc1s1* control or Rr-*Bloc1s1* cells) significantly promoted BLOS1$^+$ BCVs compared to the wt-*Bloc1s1* control (*Figure 4L*, *Figure 4—figure supplement 3E, F*). These findings demonstrated that nonfunctional BLOS1 permitted the trafficking of the pathogen from LE/Lys membranes to the ER. However, Rr-*Bloc1s1* cells (or 4µ8C-treated cells) promoted the trafficking and degradation of the pathogen in lysosomes.

## Disassembly of BORC promotes BCV trafficking to and accumulation in the ER

To test whether BORC-related lysosome trafficking components mediate BCV trafficking during infection, we analyzed the dynamics of the interaction of BCVs with LAMTOR1 (a central component of mTORC1), the small GTPase ARL8b, and kinesin KIF1b and KIF5b proteins during infection. During bacterial intracellular trafficking and replication, colocalization of BCVs with both LAMP1 and LAMTOR1, LAMP1, or LAMTOR1 decreased in control and m*Bloc1s1* cells, whereas these interactions were observed at higher levels in cells harboring Rr-*Bloc1s1* variants (*Figure 5A, B, E, G*; *Figure 4—figure supplement 3E*; *Figure 5—figure supplement 1A, B*). Similarly, recruitment of ARL8b to BCVs and/or LAMP1 was reduced in control and m*Bloc1s1* cells, which impaired their kinesin-dependent movement toward the cell periphery; however, these interactions were maintained in Rr-*Bloc1s1* cells (*Figure 5C, D, F, H*; *Figure 5—figure supplement 1C-D*). KIF1b$^+$ and KIF5b$^+$ preferentially drive lysosomes on peripheral tracks and perinuclear/ER tracks, respectively (*Guardia et al., 2016*). In control, m*Bloc1s1*, and wt-*Bloc1s1* cells, BCV interactions with KIF1b$^+$ or KIF5b$^+$ decreased (*Figure 6A, B, E, G*; *Figure 6—figure supplement 1A, B*) or increased (*Figure 6C, D, F, H*; *Figure 6—figure supplement 1C-D*), respectively. However, the opposite interaction phenomena were observed in Rr-*Bloc1s1* cells (*Figure 6*; *Figure 6—figure supplement 1*). These findings suggest that BORC-related lysosome trafficking components may regulate BCV perinuclear trafficking, fusion with ER membranes and subsequent bacterial replication.

BORC, a protein complex that contains three components (i.e., BLOS1, BLOS2, SNAPIN) shared with the BLOC-1 complex and five other proteins KXD1, C17orf59 (Lyspersin), LOH12CR1(Myrlysin), C10orf32 (Diaskedin), and MEF2BNB, plays a critical role in the regulation of lysosome positioning (*Pu et al., 2015*). In HeLa cells, interference with BORC triggers LE/Lys trafficking to the cell center via dynein, resulting in a characteristic clustering of LE/Lys in perinuclear regions (*Pu et al., 2015*). We hypothesized that degradation of *Bloc1s1* mRNA by IRE1α during *Brucella* infection interferes with BORC assembly, resulting in the alteration of recruitment or disassociation of BORC-related trafficking components, and increased LAMP1$^+$-BCV perinuclear trafficking and fusion with the ER and/or macroautophagosome membranes. To test this hypothesis, we performed protein co-immunoprecipitation (Co-IP) assays to measure the association of BORC components with each other in *Brucella*-infected or -uninfected host cells. We found that in uninfected cells, BLOS1 interacted with protein components of BLOC-1 (PALLIDIN), BORC (KXD1), and both BLOC-1 and BORC (BLOS2, SNAPIN) (*Figure 7A, B*). Moreover, under this condition, BORC components localized with peripheral or cytosolic LE/Lys membranes (*Figure 5—figure supplement 1*; *Figure 6—figure supplement 1*). However, in *Brucella*-infected cells, where *Bloc1s1* mRNA was degraded and BLOS1 protein depletion was observed (*Figure 3D, H*; *Figure 4G, H*; *Figure 3—figure supplement 3A*; *Figure 4—figure supplement 1D, E*), physical interaction between BLOS1 and BORC component SNAPIN was reduced in control cells and difficult to detect in the m*Bloc1s1* variants during intracellular trafficking and replication of the pathogen (48 h.p.i.) (*Figure 7C, D*). In fact, interactions between LYSPERSIN and KXD1 in control cells were also only detected at early time points (2 h.p.i.), but not at later time points corresponding to intervals when bacterial intracellular trafficking and replication were expected to occur; these interactions were also hardly detected in infected cells expressing m*Bloc1s1* variants at these time points (*Figure 7C, E*). The reduced interactions between BLOS1 and BORC component

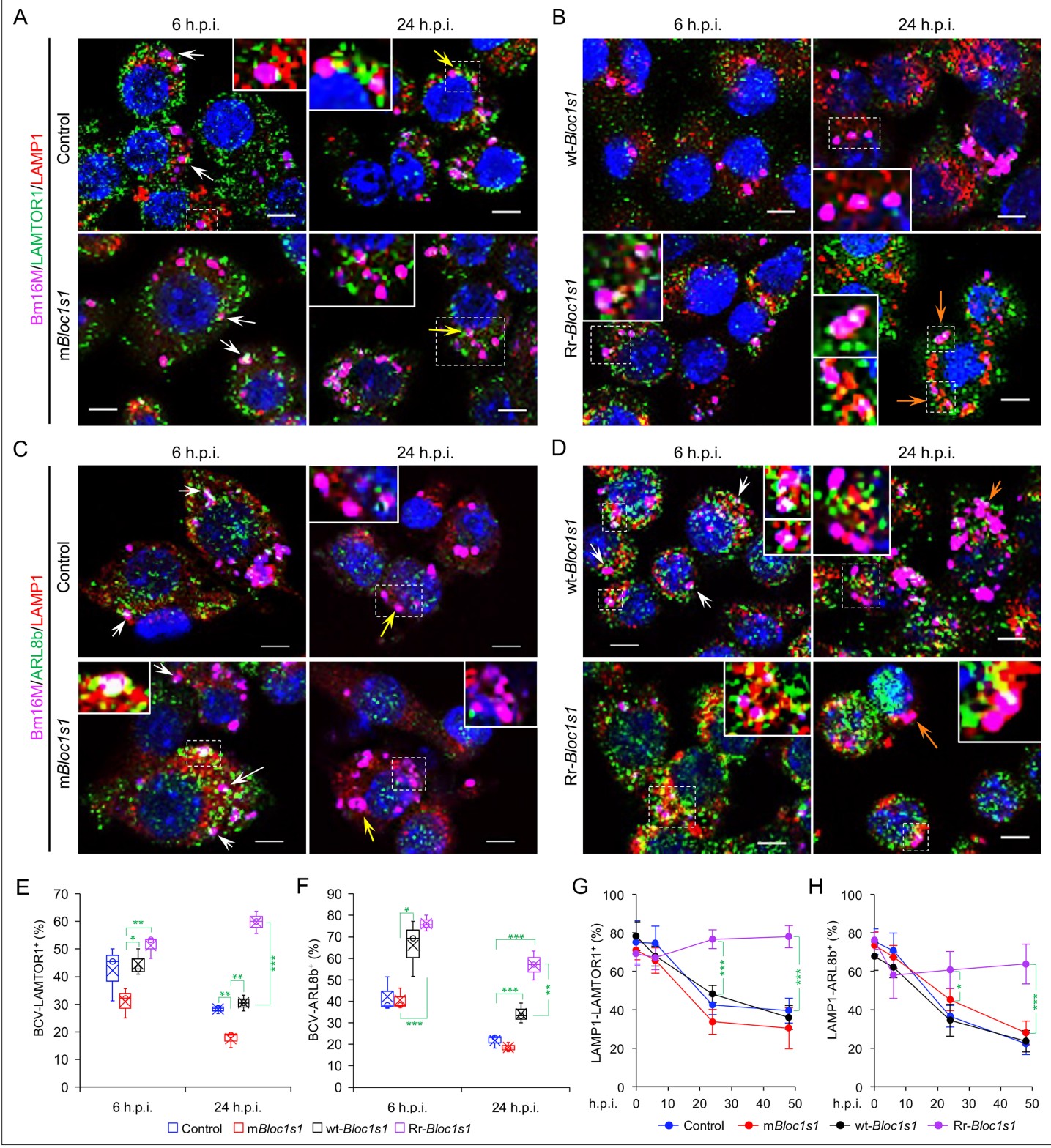

**Figure 5.** *Brucella* infection dissociates host BORC-related lysosome trafficking factor LAMTOR1 and ARL8b from lysosomes. Colocalization of LAMTOR1 with BCVs or LAMP1 in the infected control and m*Bloc1s1* (**A**), or in wt-*Bloc1s1* and Rr-*Bloc1s1* (**B**) cells at the indicated h.p.i. Colocalization of ARL8b with BCVs or LAMP1 in the infected control and m*Bloc1s1* (**C**), or in wt-*Bloc1s1* and Rr-*Bloc1s1* (**D**) cells at the indicated h.p.i. White arrows: colocalization of BCVs with the indicated proteins. Insets: magnification of the selected areas (within windows with dash white lines). Yellow and orange arrows: the perinuclear and peripheral accumulation of BCVs-Lamp1, respectively. Bar: 5 μm. Quantification of BCV-LAMTOR1⁺ (**E**) and BCV-ARL8b⁺ (**F**)

*Figure 5 continued on next page*

Figure 5 continued

in Bm16M-infected cells at the indicated h.p.i. showing in **A, B** and **C, D**, respectively. (**G, H**) Dynamics of LAMP1-LAMTOR1+ (**E**) or LAMP1-ARL8b+ (**H**) in a time course (48 hr) of Bm16M infection at the indicated h.p.i. Host cells were infected with or without Bm16M, and at the indicated h.p.i., the cells were fixed and performed confocal immunofluorescence assays. Images are representative of three independent experiments. Statistical data expressed as mean ± standard error of mean (SEM) from three independent experiments. *p < 0.05; **p < 0.01; ***p < 0.001.

The online version of this article includes the following figure supplement(s) for figure 5:

**Figure supplement 1.** Association of the indicated BORC-related lysosome trafficking components in the indicated uninfected host cells.

SNAPIN as well as LYSPERSIN and KXD1 may result from the disassembly of BORC when BLOS1 is degraded during *Brucella* infection (*Figure 7C–E*). In *Brucella*-infected *Bloc1s1* overexpressing (wt-*Bloc1s1*) cells, substantial reductions in the interactions were also observed (*Figure 7F, G*). In the infected Rr-*Bloc1s1* cells, the interaction was maintained at a relatively higher level (*Figure 7F, G*), suggesting that the BORC complex remains assembled. The integrity or disassociation of BORC was consistent with the interactions between BORC-related trafficking components and with the colocalization dynamics of BCVs with BLOS1, mTORC1/LAMP1, LAMP1/ARL8b, LAMP1/KIF1b or with LAMP1/KIF5b. These findings were also consistent with BCV peripheral or perinuclear/ER trafficking and accumulation (*Figures 5 and 6*; *Figure 4—figure supplement 3E, F*). The results collectively suggested that the degradation of *Bloc1s1* mRNA during pathogen infection resulted in the disassembly of BORC, which promoted BCV trafficking to and accumulation in the vicinities of nuclei and likely facilitated the fusion of BCVs with ER membranes in which bacteria replicated.

## Host RIDD activity on BLOS1 promotes coronavirus intracellular replication

In light of the global COVID-19 pandemic, we tested whether RIDD-controlled BLOS1 activity is a target for subversion by coronaviruses. We infected control or host cells harboring alterations in this pathway with mouse hepatitis virus [MHV, a positive-strand RNA virus classified as a member of the Betacoronavirus genus (CoV)]. Notably, previous studies have shown that MHV infection induces host cell UPR and activates IRE1α RNase and *Xbp1* splicing (*Bechill et al., 2008*), thereby suggesting the hypothesis that MHV infection of host cells activates RIDD activity. To test this hypothesis, m*Bloc1s1* or control host cells were untreated or treated with 4μ8C. Next, these cells were infected with MHV for 24 hr. Virus plaque-forming units (PFUs) and host *Bloc1s1* expression were then measured. We found that viral PFUs were reduced in 4μ8C-treated cells. However, significantly increased PFU in m*Bloc1s1* cells at 24 h.p.i. compared to controls was observed (*Figure 7H*). Expression levels of *Bloc1s1* mRNA were dramatically reduced during infection (*Figure 7I*). Collectively, these findings suggested that coronavirus MHV, like *Brucella*, subverts the host RIDD pathway to promote intracellular infection.

## Discussion

RIDD, a fundamental component of UPR in eukaryotic cells, cleaves a cohort of mRNAs encoding polypeptides that influence ER stress, thereby supporting the maintenance of ER homeostasis. In this report, we found that *Brucella* infection subverts UPR, in general (*Pandey et al., 2018*; *Qin et al., 2008*; *Smith et al., 2013*; *Taguchi et al., 2015*), and RIDD activity on *Bloc1s1*, in particular, to promote intracellular parasitism. BLOS1, encoded by RIDD gene *Bloc1s1*, is a shared subunit of both BLOC-1 and BORC complexes (*Pu et al., 2015*). Mutation or a reduction in BLOS1 expression affects both BLOC-1 and BORC (*Figure 7A–H*). The BLOC-1 complex is mainly involved in endosomal maturation and endosome–lysosome trafficking and fusion (*John Peter et al., 2013*; *Pu et al., 2015*; *Scott et al., 2018*). Our work does not rule out the possibility that the disassociation of BLOC-1 also affects *Brucella* intracellular parasitism, especially in the early stages of cellular infection.

Investigation of *Brucella*-mediated RIDD genes provides an avenue for understanding how the pathogen subverts or evades host functions to promote its intracellular lifestyle. For example, *Brucella* infection downregulates the expression of *Txnip*, a gene shown in this work to be subject to RIDD control and known to facilitate the intracellular survival of the pathogen (*Hu et al., 2020*). Cells undergoing ER stress following treatment with UPR inducers (e.g., thapsingargin, Tm) increase TXNIP protein levels and mRNA stability by reducing levels of the TXNIP destabilizing microRNA, miR-17 (*Lerner et al., 2012*). Therefore, RIDD might be expected to increase the stability of TXNIP mRNA by

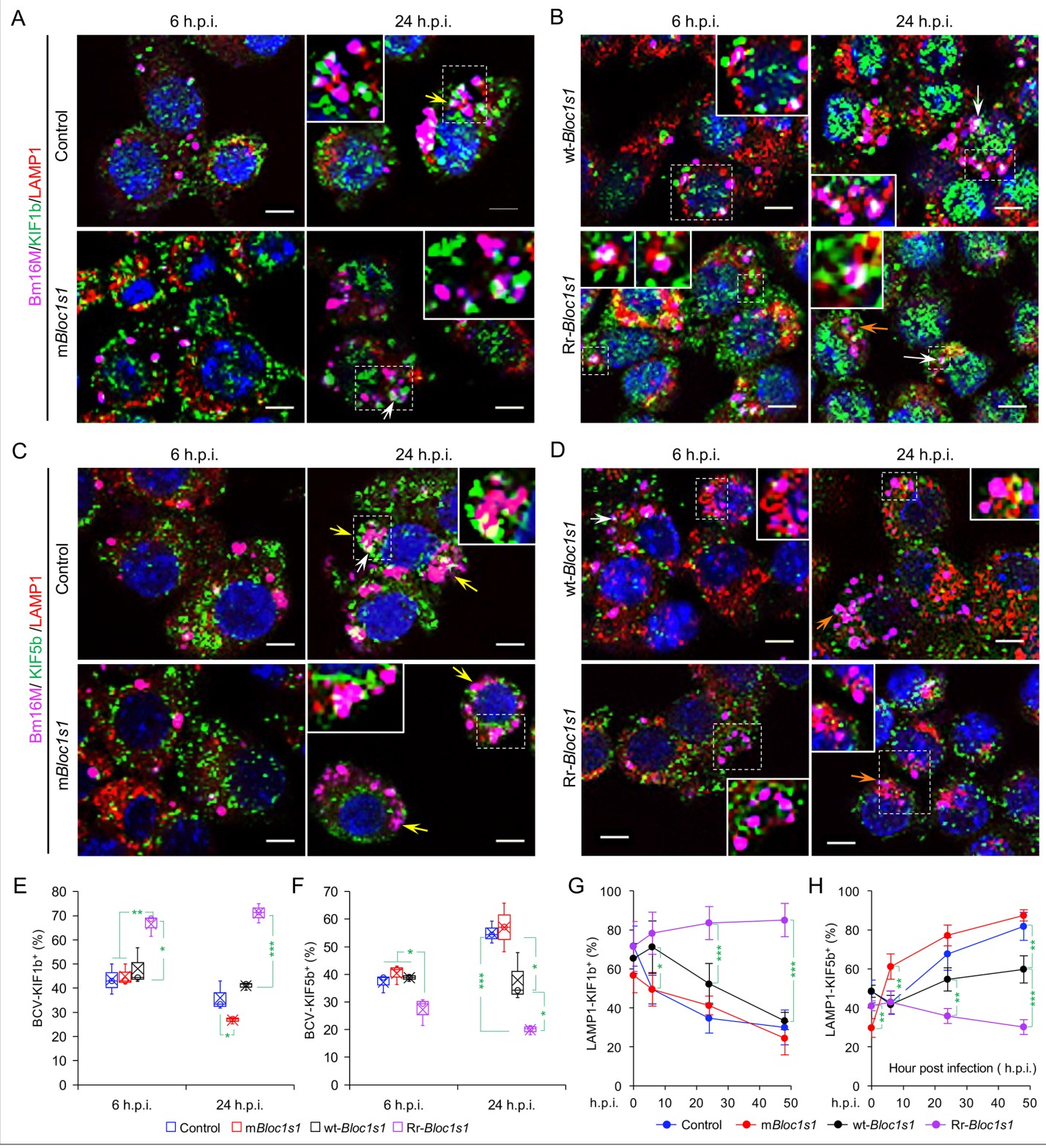

**Figure 6.** *Brucella* infection dissociates BORC-related lysosome trafficking factor KIF1b but recruits KIF5b. Colocalization of KIF1b with BCVs or LAMP1 in the infected control and m*Bloc1s1* (**A**), or in wt-*Bloc1s1* and Rr-*Bloc1s1* (**B**) cells at the indicated h.p.i. Colocalization of KIF5b with BCVs or LAMP1 in the infected control and m*Bloc1s1* (**C**), or in wt-*Bloc1s1* and Rr-*Bloc1s1* (**D**) cells at the indicated h.p.i. White arrows: colocalization of BCVs with the indicated proteins. Insets: magnification of the selected areas (within windows with dash white lines). Yellow and orange arrows: the perinuclear and peripheral accumulation of BCVs-Lamp1, respectively. Bar: 5 μm. Quantification of BCV-KIF1b+ (**E**) and BCV-KIF5b+ (**F**) in Bm16M-infected cells at the

*Figure 6 continued on next page*

*Figure 6 continued*

indicated h.p.i. showing in **A, B** and **C, D**, respectively. Dynamics of LAMP1-KIF1b$^+$ (**G**) or LAMP1-KIF5b$^+$ (**H**) in a time course (48 hr) of Bm16M infection at the indicated h.p.i. Host cells were infected with or without Bm16M, and at the indicated h.p.i., the cells were fixed and subjected to confocal immunofluorescence assays. Images are representative of three independent experiments. Statistical data represent means ± standard error of mean (SEM) from three independent experiments. *p < 0.05; **p < 0.01; ***p < 0.001.

The online version of this article includes the following figure supplement(s) for figure 6:

**Figure supplement 1.** Association of the BORC-related lysosome trafficking components KIF1b, KIF5b, and LAMP1 in the indicated uninfected host cells.

attacking the miR-17 that affects TXNIP mRNA stability. However, a recent report demonstrated that in vitro cultivated host cells or mouse tissues infected by BaS2308 display reduced expression levels of TXNIP mRNA and/or protein as well as miR-17 (*Hu et al., 2020*). Consistent with these results, we found that both TXNIP mRNA and miR-17 were destabilized during *Brucella* infection. These findings suggest that the downregulation of TXNIP during *Brucella* infection is miR-17 independent. The functions of other RIDD genes and the mechanisms by which they regulate miRNAs and their targets during *Brucella* infection constitutes an interesting area for further investigation.

Our findings support a stepwise working model by which *Brucella* subverts the host RIDD pathway to facilitate intracellular parasitism by disrupting BORC-directed lysosomal trafficking (*Figure 7J*).

First, *Brucella* enters host cell via the endocytic pathway (*Figure 7J*, right portion). Bacterial infection also induces UPR in host cells, a process associated with activation of IRE1α kinase (*Pandey et al., 2018*; *Taguchi et al., 2015*) and RNase activities (*Smith et al., 2013*; this work) (*Figure 7J*, step 1). Second, degradation of the RIDD target *Bloc1s1* by IRE1α RNase activity results in depletion of BLOS1 proteins and reduced association with BORC components (*Figure 3D* and *Figure 4G, H*; *Figure 7A–H*; *Figure 7J*, steps 2 and 3). Third, these events drive the trafficking of BCVs to the ER and their perinuclear accumulation (*Figure 7J*, steps 4 and 5), mitigate further fusion of BCVs with cytosolic lysosomes, and limit BCV trafficking to LE/Lys in peripheral regions where these organelles possess enhanced degradative functions (*Figures 5 and 6*; *Figure 4—figure supplement 3E, F*; *Figure 7J*, dash-blue line). Finally, the accumulation of BCVs decorated with ER proteins increases due to the fusion of BCVs with ER membranes and/or with noncanonical macrophagosomes (*Pandey et al., 2018*; *Starr et al., 2012*; *Taguchi et al., 2015*; *Figure 7J*, steps 5 and 6). These final events support the intracellular replication of the pathogen (*Figure 7J*).

Several lines of evidence support the proposed mechanism. First, *Brucella* infection activates IRE1α RNase activity as evidenced by *Xbp1* mRNA splicing (*Figure 3—figure supplement 1A*). However, the intracellular replication of the pathogen is not impaired in *Xbp1* KO cells and mice (*Figure 1N–P*). These findings support the hypothesis that IRE1α RNase activity is required for *Brucella* infection in an IRE1α-XBP1 independent fashion. Second, in addition to splicing *Xbp1*, IRE1α cleaves other mRNAs, resulting in their RIDD-mediated decay (*Bae et al., 2019*). We identified several mRNAs, including *Bloc1s1* (*Figure 3A–C*), that contain predicted stem-loop structures that were inferred to be targets of IRE1α RNase activity (*Moore and Hollien, 2015*). The expression of these mRNAs was downregulated in response to *Brucella* infection. Host cells harboring nonfunctional *Bloc1s1* mutants were highly susceptible to pathogen infection, whereas cells that express a RIDD-resistant *Bloc1s1* variant were resistant to *Brucella* infection (*Figure 4E, F*).

Third, lysosome positioning regulated by BORC is a critical determinant of its functions. BORC associates peripherally with lysosomal membranes, where it recruits the small GTPase ARL8b to lysosomes. BORC and ARL8b promote lysosome movement by coupling to kinesin-1 (KIF5b) or kinesin-3 (KIF1b), which preferentially moves lysosomes on perinuclear tracks enriched in acetylated α-tubulin or on peripheral tracks enriched in tyrosinated α-tubulin, respectively (*Guardia et al., 2016*; *Pu et al., 2015*). Interference with BORC or other components of this pathway drives lysosome trafficking to the cell center via dynein. Thus, cells lacking BORC display a perinuclear clustering of lysosomes (*Pu et al., 2015*). Ragulator (a GEF for the Rag GTPases that signal amino acid levels to mTORC1) directly interacts with and inhibits BORC functions (*Pu et al., 2017*). Building upon these observations, we show that *Brucella* infection results in *Bloc1s1* degradation and disassembly of BORC; moreover, during *Brucella* intracellular trafficking and replication, colocalization of mTORC1, ARL8b, and KIF1b with BCVs or lysosomes was reduced in control cells, and in cells expressing nonfunctional variants of BLOS1; however, KIF5b localization with BCVs or lysosomes was increased or in a higher level in these

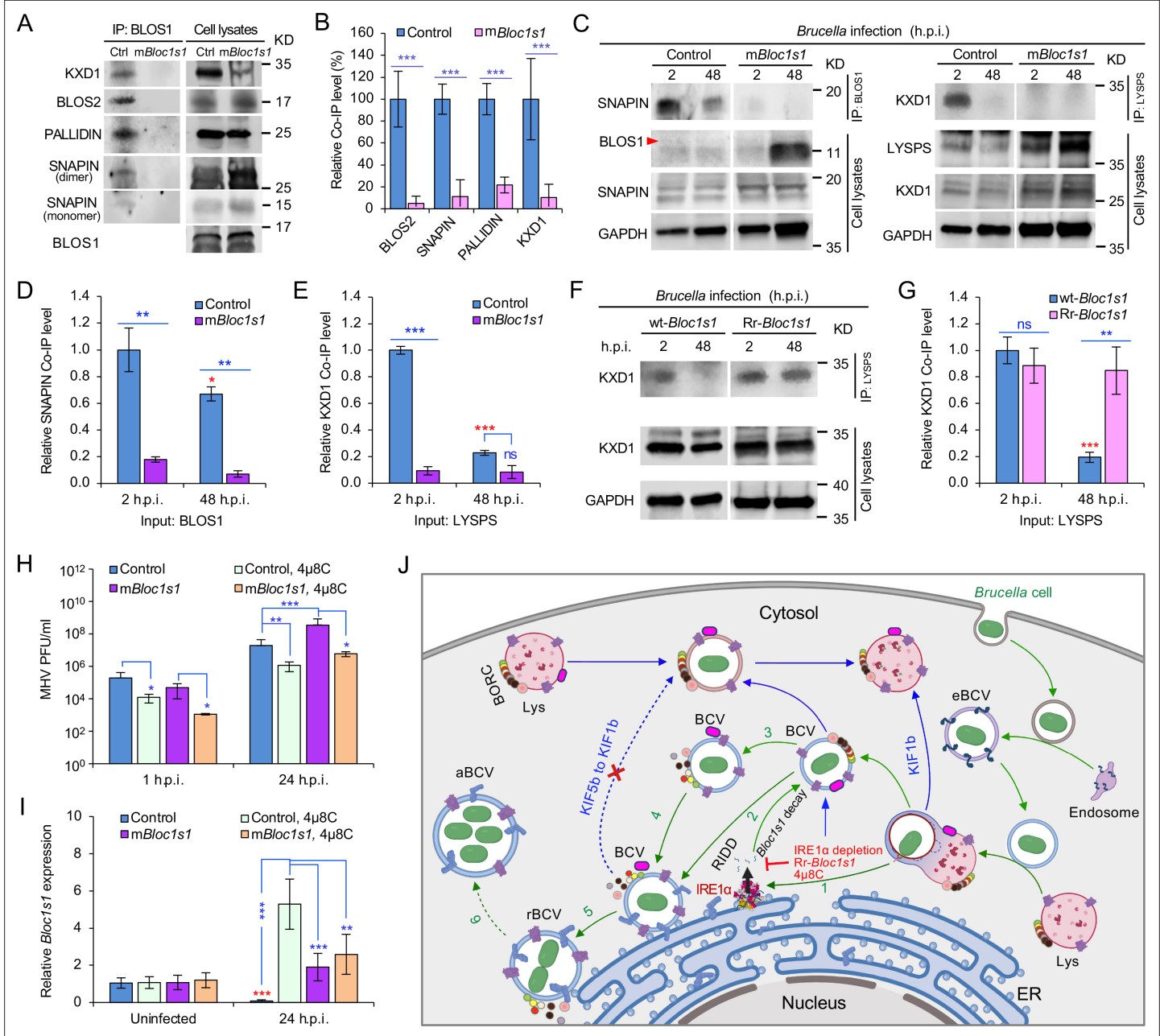

**Figure 7.** *Brucella* infection dissociates host BORC and degradation of *Bloc1s1* mRNA supports coronavirus intracellular replication. (**A**) Co-immunoprecipitation (Co-IP) analysis of the interactions of BLOS1 with other proteins that form the BLOC-1 (BLOS1, BLOS2, SNAPIN, and PALLIDIN) or BORC (BLOS1, BLOS2, SNAPIN, and KXD1) complex. The nonfunctional m*Bloc1s1* and control cells were cultured overnight before being subjected to Co-IP assays with BLOS1 as an input. (**B**) Quantification of the indicated pulled-down protein levels from overnight-cultured control and m*Bloc1s1* cell lysates using BLOS1 as an input. (**C**) Co-IP assays for *Brucella*-infected control and m*Bloc1s1* cells at the indicated h.p.i. using BLOS1 (left panel) or LYSPERSIN (LYSPS, right panel) as an input. Red arrow: BLOS1. Quantification of the indicated pulled-down protein levels of SNAPIN (**D**) or KXD1 (**E**) from *Brucella*-infected control and m*Bloc1s1* cell lysates using BLOS1 or LYSPS as an input. Red asterisks: significance when compared to the control at 2 h.p.i. (**F**) Co-IP assays for *Brucella*-infected wt- or Rr-*Bloc1s1* cells at the indicated h.p.i. using LYSPERSIN as an input. (**G**) Quantification of the indicated pulled-down protein levels from *Brucella*-infected cell lysates of wt- or Rr-*Bloc1s1*. Red asterisk: significance when compared to the wt-*Bloc1s1* control at 2 h.p.i. (**H**) PFU (plaque-forming units) assay of coronavirus MHV infection of the m*Bloc1s1* and control cells treated or untreated with 4μ8C (50 μM) at the indicated h.p.i. (**I**) *Bloc1s1* mRNA expression assay of the indicated host cells infected with MHV via qRT-PCR. Red asterisk: significance when compared to the uninfected control. (**J**) A proposed model describing how *Brucella* subverts the host RIDD-BLOS1 pathway to support intracellular parasitism by disrupting BORC-directed lysosomal trafficking. BCV: *Brucella*-containing vacuole. eBCV, rBCV, and aBCV: endosomal BCV, replicative BCV, and autophagic BCV, respectively. BORC: the BLOC-1-related complex. Green arrows: BCV trafficking to the ER compartment and replication. Blue arrows: BCV trafficking to peripheral lysosome and lysosomal degradation. Host cells were infected with or without *Brucella* or MHV, and at the

*Figure 7 continued on next page*

*Figure 7 continued*

indicated h.p.i., the infected or uninfected host cells were harvested for Co-IP and immunoblotting assays, or qRT-PCR assays. Blots are representative of three independent experiments. Statistical data represent the mean ± standard error of mean (SEM) from three independent experiment. *p < 0.05; **p < 0.01; ***p < 0.001. Red asterisks: Compared to the same *Brucella*-infected cells at 2 h.p.i.

cells (*Figure 5*, *Figure 5—figure supplement 1*; *Figure 6*, *Figure 6—figure supplement 1*). Colocalization of the BORC-related lysosome trafficking factors (i.e., ARL8b, KIF1b, and mTORC1) with BCVs or lysosomes in cells expressing Rr-*Bloc1s1* variants were maintained at a relatively higher level (*Figures 5 and 6*). These findings demonstrate that blocking BORC function via the disassembly of the BORC complex through depletion of *Bloc1s1* by *Brucella* infection drives BCVs toward the perinuclear region and ER accumulation, which likely facilitates the fusion of BCVs with the ER, thereby supporting intracellular parasitism.

Finally, RIDD-mediated *Bloc1s1* degradation may promote BCV fusion with autophagosomes. Nutrient-starved cells display perinuclear clustering of lysosomes, which influences autophagosome formation and autophagosome–lysosome fusion rates (*Korolchuk et al., 2011*). Lysosome perinuclear clustering during starvation, ER stress induced by accumulation of misfolded proteins, drug treatments, and pathogen infection can disrupt metabolic homeostasis, thereby necessitating the induction of cell biological processes that return the cell to equilibrium. Macroautophagy and degradation of sequestered cytosolic materials by fusion of autophagosomes/macrophagosomes with lysosomes can promote the re-establishment of homeostasis (*Bae et al., 2019*; *Korolchuk et al., 2011*; *Pu et al., 2017*). Degradation of *Bloc1s1* mRNA by IRE1α leads to the perinuclear accumulation of LE/Lys in response to ER stress in mouse cells. Overriding *Bloc1s1* degradation results in ER stress sensitivity and the aggregation of ubiquitinated proteins. The LE/Lys perinuclear-trafficking and LE-associated endocytic transport promote the efficient degradation of these protein aggregates. Therefore, *Bloc1s1* regulation via RIDD facilitates LE-mediated autophagy of protein aggregates, thereby promoting cell survival during stress (*Bae et al., 2019*). Hepatocytes from *Bloc1s1* liver-specific knockout (*DelVecchio et al., 2002*) cells accumulate autolysosomes and lysosomes. In LKO hepatocytes, the initiation or extension of lysosomal tubules is abolished, which impairs autophagic lysosome reformation and results in the accumulation of enlarged autolysosomes (*Wu et al., 2021b*). *Bloc1s1* degradation by the RIDD pathway promotes BCV perinuclear or ER-region clustering and may also avoid the peripheral movement of BCVs away from the ER region as a consequence of reduced α-tubulin acetylation. These processes may facilitate BCV fusion with ER membranes or (macro)phagosomes, promote the enlargement of aBCVs and further bacterial replication, and ultimately relieve *Brucella*-induced ER stress (*Pandey et al., 2018*; *Qin et al., 2008*; *Starr et al., 2012*; *Taguchi et al., 2015*).

In addition to *Brucella*, the betacoronavirus MHV also subverts the RIDD–BLOS1 axis to promote intracellular replication (*Figure 7H, I*), thereby indicating that RIDD control of BLOS1 activity is not pathogen specific. How the host RIDD–*Bloc1s1* axis regulates interactions between host cells and coronaviruses merits further investigation. However, additional possibilities for regulatory control can be envisioned. First, coronaviruses utilize many proteins such as nsp1 to inhibit host protein synthesis in the first 6 hr of infection (*Nakagawa and Makino, 2021*). Second, BLOS1 contains a potential coronavirus 3C-like protease cleavage site, LQ^SAPS, near its C-terminus, thereby rendering it potentially susceptible to direct subversion by coronaviral pathogens. Finally, coronaviruses have evolved to subvert host interferon defenses (*Thoms et al., 2020*), which may contribute to immune evasion. Future work will be directed toward examining these possibilities and the roles and mechanisms by which the RIDD-*Bloc1s1* axis controls these and other host–pathogen interactions.

## Materials and methods

All the key resources used in the work are listed in Key resources table used in this work.

### Key resources table

| Reagent type (species) or resource | Designation | Source or reference | Identifiers | Additional information |
|---|---|---|---|---|
| Strain, strain background (*Brucella abortus*) | Strain 2308 | Maintained by the de Figueiredo lab | | WT strain |

*Continued on next page*

*Continued*

| Reagent type (species) or resource | Designation | Source or reference | Identifiers | Additional information |
|---|---|---|---|---|
| Strain, strain background (*B. abortus*) | S19 | Maintained by the de Figueiredo lab | | Smooth vaccine strain |
| Strain, strain background (*B. melitensis*) | 16M | Maintained by the de Figueiredo lab | | WT strain |
| Strain, strain background (*B. melitensis*) | 16M Δ*VirB2* | Maintained by the de Figueiredo lab | | Δ*VirB2* derived from 16M |
| Strain, strain background (*B. melitensis*) | 16M Δ*VjbR* | Maintained by the de Figueiredo lab | | Δ*vjbR* derived from 16M |
| Strain, strain background (Mouse Hepatitis Virus) | A59 | Other | | Gift from Leibowitz lab |
| Strain, strain background (Lentivirus) | LLV-LentiGUIDE-Puro-Bloc1s1.1-gRNA | This paper | | The de Figueiredo lab; see *Methods and methods* |
| Strain, strain background (Lentivirus) | LLV-LentiGUIDE-Puro-Bloc1s1.2-gRNA | This paper | | The de Figueiredo lab; see *Methods and methods* |
| Strain, strain background (Lentivirus) | LLV-LentiGUIDE-Puro-Bloc1s1.3-gRNA | This paper | | The de Figueiredo lab; see *Methods and methods* |
| Strain, strain background (Lentivirus) | LLV-LentiGUIDE-Puro-Bloc1s1.4-gRNA | This paper | | The de Figueiredo lab; see *Methods and methods* |
| Strain, strain background (Lentivirus) | LLV-LentiGUIDE-Puro-Bloc1s1.5-gRNA | This paper | | The de Figueiredo lab; see *Methods and methods* |
| Strain, strain background (Lentivirus) | LLV-LentiGUIDE-Puro-GFP2 | Other | | Gift from the Watson/Patrick lab; Replication incompetent lentivirus |
| Strain, strain background (Lentivirus) | LLV-Lenti-CMV-Hygro-DEST (w117-1)-Bloc1s1-WT | This paper | | The de Figueiredo lab; see *Methods and methods* |
| Strain, strain background (Lentivirus) | LLV-Lenti-CMV-Hygro-DEST (w117-1)-Bloc1s1-G449T | This paper | | The de Figueiredo lab; see *Methods and methods* |
| Strain, strain background (*Escherichia coli*) | MAX Efficiency DH5α Competent Cells | Thermo Fisher Scientific | Cat# 18258012 | |
| Strain, strain background (*Escherichia coli*) | Stbl3 | Other | | Gift from the Watson/Patrick lab; Chemically competent cells |
| Strain, strain background (*Escherichia coli*) | XL10 Gold | Aligent Technologies | Cat# 210,518 | From QuikChange Lightning Site-Directed Mutagenesis Kit |
| Strain, strain background (*Escherichia coli*) | DB3.1 | **Campeau et al., 2009** | Addgene plasmid #17,454 | Containing pLenti CMV Hygro Dest (w117-1) |
| Genetic reagent (*Mus musculus*) | *Lyz2-Ern1*$^{wt/wt}$ | **Iwawaki et al., 2009** | | Gift from the Iwawaki lab |
| Genetic reagent (*Mus musculus*) | *Ern1*$^{mut/mut}$; *Lyz2-Cre* (*Ern1* CKO) | **Iwawaki et al., 2009** | | Gift from the Iwawaki lab |
| Genetic reagent (*Mus musculus*) | *Xbp1*$^{+/+}$ | Other | | Gift from Dr. Laurie Glimcher |
| Genetic reagent (*Mus musculus*) | *Xbp1*$^{-/-}$ | Other | | Gift from Dr. Laurie Glimcher |
| Cell line (*Mus musculus*) | J774A.1 | ATCC | Cat# TIB-67; RRID:CVCL_0358 | Cells tested negative for mycoplasma; cells were maintained in cell culture media containing penicillin/streptomycin to prevent mycoplasma contamination. The cells are maintained in the Fitch lab. |
| Cell line (*Mus musculus*) | *Ern1*$^{+/+}$ MEFs | Other | | Gift from the Kaufman Lab; cells tested negative for mycoplasma; cells were maintained in cell culture media containing penicillin/streptomycin to prevent mycoplasma contamination. |
| Cell line (*Mus musculus*) | *Ern1*$^{-/-}$ MEFs | Other | | Gift from the Kaufman Lab; cells tested negative for mycoplasma; cells were maintained in cell culture media containing penicillin/streptomycin to prevent mycoplasma contamination. |
| Cell line (*Mus musculus*) | RAW 264.7 | ATCC | Cat# TIB-71; RRID:CVCL_0493 | Cells tested negative for mycoplasma; cells were maintained in cell culture media containing penicillin/streptomycin to prevent mycoplasma contamination. |

*Continued*

| Reagent type (species) or resource | Designation | Source or reference | Identifiers | Additional information |
|---|---|---|---|---|
| Cell line (*Mus musculus*) | MC3T3-E1 | *Bae et al., 2019* | | Gift from the Hollien lab; cells tested negative for mycoplasma; cells were maintained in cell culture media containing penicillin/streptomycin to prevent mycoplasma contamination. |
| Cell line (*Mus musculus*) | L2 | Other | | Gift from the Leibowitz lab; this cell line is different from the L2 cell line in ATCC; this cell line can only be obtained from MHV researchers; cells tested negative for mycoplasma; cells were maintained in cell culture media containing penicillin/streptomycin to prevent mycoplasma contamination. |
| Transfected construct (*Mus musculus*) | Raw 264.7 cell line: Cas9 | Other | | Gift from the Watson/Patrick lab; cells tested negative for mycoplasma; cells were maintained in cell culture media containing penicillin/streptomycin to prevent mycoplasma contamination. |
| Transfected construct (*Mus musculus*) | Raw 264.7 cell line: Cas9 m*Bloc1s1* | This paper | | The de Figueiredo lab; see *Methods and methods*; cells were maintained in cell culture media containing penicillin/streptomycin to prevent mycoplasma contamination. |
| Transfected construct (*Mus musculus*) | Raw 264.7 cell line: Cas9 Control | This paper | | The de Figueiredo lab; see *Methods and methods*; cells were maintained in cell culture media containing penicillin/streptomycin to prevent mycoplasma contamination. |
| Transfected construct (*Mus musculus*) | Raw 264.7 cell line: wt-*Bloc1s1* | This paper | | The de Figueiredo lab; see *Methods and methods*; cells were maintained in cell culture media containing penicillin/streptomycin to prevent mycoplasma contamination. |
| Transfected construct (*Mus musculus*) | Raw 264.7 cell line: RIDD-resistant *Bloc1s1* (Rr-*Bloc1s1*) | This paper | | The de Figueiredo lab; see *Methods and methods*; cells were maintained in cell culture media containing penicillin/streptomycin to prevent mycoplasma contamination. |
| Transfected construct (*Mus musculus*) | MC3T3-E1 cell line: RFP expressing | *Bae et al., 2019* | | Gift from the Hollien lab; cells tested negative for mycoplasma; cells were maintained in cell culture media containing penicillin/streptomycin to prevent mycoplasma contamination. |
| Transfected construct (*Mus musculus*) | MC3T3-E1 cell line: Blos1$^s$-Flag (similar to Rr-*Bloc1s1*) | *Bae et al., 2019* | | Gift from the Hollien lab; cells tested negative for mycoplasma; cells were maintained in cell culture media containing penicillin/streptomycin to prevent mycoplasma contamination. |
| Biological sample (*Mus musculus*) | Bone marrow-derived macrophages | This paper | | The de Figueiredo lab; see *Methods and methods*; cells were maintained in cell culture media containing penicillin/streptomycin to prevent mycoplasma contamination. |
| Antibody | IRE1α antibody | Novus Biologicals; Thermo Fisher Scientific | Cat: NB100-2324; PA5-20189 | WB (1:500–1000) |
| Antibody | Phospho IRE1α antibody | GeneTex Inc; Thermo Fisher Scientific; Abcam | Cat #: GTX63722; PA1-16927; ab48187 | WB (1:500–1000) |
| Antibody | Rat polyclonal LAMP1 Antibody (1D4B) | Santa Cruz Biotechnology | Cat# sc-19992 | IF (1:500) |
| Antibody | Rabbit polyclonal MAP LC3β (H-50) | Santa Cruz Biotechnology | Cat# sc-28266 | IF (1:300) |
| Antibody | Goat polyclonal Cathepsin D Antibody (R-20) | Santa Cruz Biotechnology | Cat# sc-6487 | IF (1:200) |
| Antibody | Rabbit polyclonal Cathepsin D (H-75) | Santa Cruz Biotechnology | Cat# sc-10725 | IF (1:200) |
| Antibody | Goat polyclonal Calregulin (N-19) | Santa Cruz Biotechnology | Cat# sc-6468 | IF (1:300) |
| Antibody | Rabbit polyclonal Calregulin (H-170) | Santa Cruz Biotechnology | Cat# sc-11398 | IF (1:300) |
| Antibody | alpha Tubulin Rabbit Polyclonal Antibody | Thermo Fisher Scientific | Cat# PA5-29444 | IF (1:1000) |
| Antibody | Acetyl-α-Tubulin (Lys40) (D20G3) XP Rabbit mAb | Cell signaling technology | Cat# 5335 | WB (1:1000) |
| Antibody | Rabbit polyclonal GAPDH Antibody (FL-335) HRP | Santa Cruz Biotechnology | Cat# sc-25778 HRP | WB (1:500) |
| Antibody | Goat polyclonal anti-Brucella | BEI | Cat# DD-17 AB-G-BRU-M | IF (1:400) |
| Antibody | Brucella Rabbit Polyclonal Antibody | Bioss Antibodies | Cat# bs-2229R | IF (1:1000) |
| Antibody | Chicken serum anti-Blos1 | This paper | | The de Figueiredo lab; Chicken Serum IF (1:300), WB (1:500) |

*Continued on next page*

*Continued*

| Reagent type (species) or resource | Designation | Source or reference | Identifiers | Additional information |
|---|---|---|---|---|
| Antibody | Anti-SNAPIN Rabbit Polyclonal Antibody | Proteintech | Cat# 10055-1-AP | WB (1:700) |
| Antibody | Anti-PLDN Rabbit Polyclonal Antibody | Proteintech | Cat# 10891-2-AP | WB (1:800) |
| Antibody | Anti-KIF1B Rabbit Polyclonal Antibody | Proteintech | Cat# 15263-1-AP | IF (1:200) |
| Antibody | Anti-KIF5B Rabbit Polyclonal Antibody | Proteintech | Cat# 21632-1-AP | IF (1:200) |
| Antibody | Rabbit Polyclonal IHCPlus ARL8B Antibody aa72-121 Polyclonal IHC, WB LS-B5831 | LifeSpan Biosciences | Cat# LS-B5831-100 | 5 µg/ml dilution |
| Antibody | Rabbit polyclonal C17orf59 Antibody (aa186-215) LS-C167955 | LifeSpan Biosciences | Cat# LS-C167955-400 | WB (1:1000) |
| Antibody | Rabbit polyclonal mTOR Antibody | Cell Signaling Techology | Cat# 2972 S | IF (1:300) |
| Antibody | Rabbit polyclonal KxDL motif containing 1 Antibody | Novus Biologicals | Cat# NBP1-82055 | WB (1:500) |
| Antibody | Rabbit polyclonal BLOS2 Polyclonal Antibody | Thermo Fisher Scientific | Cat# PA525452 | WB (1:1000) |
| Antibody | Alexa Fluor 488 Donkey anti-Rat IgG (H + L) | Thermo Fisher Scientific | Cat# A21208 | IF (1:300) |
| Antibody | Alexa Fluor 488 Donkey anti-Rabbit IgG (H + L) | Thermo Fisher Scientific | Cat# A21206 | IF (1:500) |
| Antibody | Alexa Fluor 488 Chicken anti-Goat IgG (H + L) | Thermo Fisher Scientific | Cat# A21467 | IF (1:500) |
| Antibody | Alexa Fluor 488 Chicken anti-Rabbit IgG (H + L) | Thermo Fisher Scientific | Cat# A21441 | IF (1:300) |
| Antibody | Alexa Fluor 594 Donkey anti-Goat IgG (H + L) | Thermo Fisher Scientific | Cat# A11058 | IF (1:500) |
| Antibody | Donkey anti-Rabbit IgG (H + L) ReadyProbes Secondary Antibody, Alexa Fluor 594 | Thermo Fisher Scientific | Cat# R37119 | 1 drop |
| Antibody | Donkey Anti-Chicken IgY Antibody (Alexa Fluor 594) | Jackson ImmunoResearch | Cat# 703-585-155 | IF (1:600) |
| Antibody | Donkey anti Rat IgG (H + L) Secondary Antibody, Alexa Fluor 594, Invitrogen | Thermo Fisher Scientific | Cat# A21209 | IF (1:300) |
| Antibody | Alexa Fluor 594 Goat anti-Rabbit IgG (H + L) | Thermo Fisher Scientific | Cat# A11012 | IF (1:300) |
| Antibody | Alexa Fluor 647 Donkey anti-Goat IgG (H + L) | Thermo Fisher Scientific | Cat# A21447 | IF (1:500) |
| Antibody | Donkey anti Chicken IgY Secondary Antibody, HRP, Invitrogen | Thermo Fisher Scientific | Cat# SA172004 | WB (1:15000) |
| Antibody | Goat anti-Rabbit IgG HRP | Santa Cruz Biotechnology | Cat# sc-2004 | WB (1:2000) |
| Recombinant DNA reagent | (plasmids) See *Supplementary file 1* | Other and this paper | | See *Supplementary file 1* Recombinant plasmids constructed in this work |
| Sequence-based reagent | (oligonucleotides) See *Supplementary file 2* | This paper | | The de Figueiredo lab; See *Supplementary file 2* |
| Peptide, recombinant protein | Blos1 N-terminal biotinylated | Genscript | Cat# SC1848 | Peptide synthesized by Genscript, used to make the Chicken Serum Blos1 antibody |
| Commercial assay or kit | Phagocytosis Assay Kit (IgG FITC) | Cayman Chemical | Cat# 500,290 | |
| Commercial assay or kit | QuikChange Lightning Site-Directed Mutagenesis Kit | Aligent Technologies | Cat# 210,518 | |

*Continued on next page*

*Continued*

| Reagent type (species) or resource | Designation | Source or reference | Identifiers | Additional information |
|---|---|---|---|---|
| Commercial assay or kit | Thermo Scientific Pierce Co Immunoprecipitation Kit | Thermo Fisher Scientific | Cat# 26,149 | |
| Commercial assay or kit | iScript Select cDNA Synthesis Kit | Bio-Rad | Cat# 1708896 | |
| Commercial assay or kit | Applied Biosystems High-Capacity cDNA Reverse Transcription Kit | Thermo Fisher Scientific | Cat# 4368814 | |
| Commercial assay or kit | RNeasy Plus Mini Kit | Qiagen | Cat# 74,134 | |
| Commercial assay or kit | TOPO TA Cloning Kit for Subcloning, without competent cells | Thermo Fisher Scientific | Cat# 451,641 | |
| Commercial assay or kit | Gateway LR Clonase II Enzyme mix | Thermo Fisher Scientific | Cat# 11791020 | |
| Chemical compound, drug | 4μ8c | Sigma-Aldrich | Cat# SML0949 | 50 μM dose |
| Chemical compound, drug | Tunicamycin from Streptomyces | Sigma-Aldrich | Cat# T7765 | 5 μg/ml dose |
| Chemical compound, drug | SsoAdvanced Universal SYBR Green Supermix | BioRad | Cat# 1725274 | |
| Chemical compound, drug | SuperBlock T20 (TBS) Blocking Buffer | Thermo Fisher Scientific | Cat# 37,536 | |
| Chemical compound, drug | Western Blot Strip-It Buffer | Advansta | Cat# R-03722-D50 | |
| Chemical compound, drug | Tween 20 | Fisher scientific | Cat# BP337-500 | |
| Chemical compound, drug | 4% wt/vol Formaldehyde made from paraformaldehyde powder | Sigma-Aldrich | Cat# 158,127 | |
| Chemical compound, drug | Prestained Protein Ladder – Extra broad molecular weight (5–245 kDa) | Abcam | Cat# ab116029 | |
| Chemical compound, drug | SuperSignal West Femto Maximum Sensitivity Substrate | Thermo Fisher Scientific | Cat# 34,095 | |
| Chemical compound, drug | RIPA Buffer | Sigma-Aldrich | Cat# RO278 | |
| Chemical compound, drug | Phosphatase inhibitor cocktail 2 | Sigma-Aldrich | Cat# P5726 | |
| Chemical compound, drug | Phosphatase inhibitor cocktail 3 | Sigma-Aldrich | Cat# P0044 | |
| Chemical compound, drug | TaqMan Universal PCR Master Mix | Thermo Fisher Scientific | Cat# 1804045 | |
| Chemical compound, drug | Puromycin | Invivogen | Cat# ant-pr-1 | |
| Chemical compound, drug | Blasticidin | Invivogen | Cat# ant-bl-1 | |
| Chemical compound, drug | Hygromycin | Invivogen | Cat# ant-hg-1 | |
| Chemical compound, drug | Gram Crystal Violet | Becton Dickinson | Cat# 212,525 | |
| Chemical compound, drug | ProLong Glass Antifade Mountant with NucBlue Stain | Thermo Fisher Scientific | Cat# P36981 | |
| Software, algorithm | ImageJ | https://imagej.nih.gov/ij/ | RRID:SCR_003070 | Version 1.52t (64 bit) |
| Software, algorithm | Fiji | https://imagej.net/Fiji | RRID:SCR_002285 | Version 2.3.0/1.53f51 (64 bit) |
| Software, algorithm | Coloc2 | https://imagej.net/Coloc_2 | | |
| Software, algorithm | GraphPad Prism version 8.2.0 for Windows | GraphPad Software | RRID:SCR_002798 | https://www.graphpad.com/scientific-software/prism/ |
| Software, algorithm | Gen5 | BioTek | | Version 3.05 |

*Continued on next page*

*Continued*

| Reagent type (species) or resource | Designation | Source or reference | Identifiers | Additional information |
|---|---|---|---|---|
| Software, algorithm | ICE analysis | Synthego, (*Synthego Performance Analysis, 2019*) | | |
| Software, algorithm | RNAfold WebServer | University of Vienna | RRID:SCR_008550 | http://rna.tbi.univie.ac.at/cgi-bin/RNAWebSuite/RNAfold.cgi |
| Software, algorithm | SnapGene | GSL Biotech LLC | SCR_015052 | https://www.snapgene.com/ |
| Software, algorithm | STAR version 2.5.2a | *Dobin et al., 2013* | RRID:SCR_004463 | STAR: ultrafast universal RNA-seq aligner; http://chagall.med.cornell.edu/RNASEQcourse/STARmanual.pdf |
| Software, algorithm | RSEM version 1.2.29 | *Li and Dewey, 2011* | RRID:SCR_013027 | RSEM: accurate transcript quantification from RNA-Seq data with or without a reference genome; https://deweylab.github.io/RSEM/ |
| Software, algorithm | DESeq version 1.30.0 | *Anders and Huber, 2010* | RRID:SCR_000154 | Differential expression analysis for sequence count data; https://www.bioconductor.org/packages//2.10/bioc/html/DESeq.html |
| Software, algorithm | WebGestalt-2013 | *Wang et al., 2013* | RRID:SCR_006786 | WEB-based GEne SeT AnaLysis Toolkit (WebGestalt): update 2013; http://www.webgestalt.org |
| Software, algorithm | PANTHER version 7 | *Gaudet et al., 2011*; *Mi et al., 2010* | RRID:SCR_004869 | Phylogenetic-based propagation of functional annotations within the Gene Ontology consortium; http://www.pantherdb.org |
| Software, algorithm | edgeR version 3.26.6 | *McCarthy et al., 2012*; *Robinson et al., 2010* | RRID:SCR_012802 | https://www.bioconductor.org/packages/release/bioc/html/edgeR.html |
| Software, algorithm | Cytoscape | *Zhao et al., 2017* | RRID:SCR_003032 | https://cytoscape.org/ |
| Software, algorithm | Biorender | Biorender | RRID:SCR_018361 | https://biorender.com/ |
| Other | SeaPlaqueTM Agarose | Lonza | Cat# 50,101 | Used for plaque assays |
| Other | Accell Mouse Kif5b siRNA SMARTpool 5 nmol | Horizon | Cat# E-040710-00-0005 | Used for validation |
| Other | Accell Mouse Kif1b siRNA SMARTpool 5 nmol | Horizon | Cat# E-040900-00-0005 | Used for validation |
| Other | Accell Mouse Arl8b siRNA SMARTpool 5 nmol | Horizon | Cat# E-056525-00-0005 | Used for validation |
| Other | Accell Non-targeting Control Pool 5 nmol | Horizon | Cat# D-001910-10-05 | Used for validation |
| Other | Accell siRNA Delivery Media | Horizon | Cat# B-005000-500 | Used for validation |
| Other | ON-TARGETplus Human BLOC1S1 (2647) siRNA - SMARTpool, 5 nmol | Dharmacon | L-012580-01-0005 | Used for validation |
| Other | 8–16% Mini-PROTEAN TGX Precast Protein Gels, 10-well, 30 µl | BioRad | 4561103 | |
| Other | DAPI | Sigma-Aldrich | Cat# D9542-10MG | 1 µg/ml |

## Bacterial strains, cell culture, *Brucella* infection, and antibiotic protection assays

*Brucella melitensis* strain 16M (WT), and *B. abortus* strain 2308 (WT), and *B. abortus* vaccine strain S19 and other bacterial strains were used in this work. Bacteria were grown in tryptic soy broth (TSB) or on tryptic soy agar (TSA, Difco) plates, supplemented with either kanamycin (Km, 50 µg/ml) or chloramphenicol (Cm, 25 µg/ml) when required. For infection, 4 ml of TSB was inoculated with a loop of bacteria taken from a single colony grown on a freshly streaked TSA plate. Cultures were then grown with shaking at 37°C overnight, or until $OD_{600} \approx 3.0$.

Mammalian host cells including murine macrophages RAW264.7 and its derived nonfunctional and Rr-*Bloc1s1* variants and corresponding control cells, BMDMs, J774.A1 cells, and MEFs were routinely cultured at 37°C in a 5% $CO_2$ atmosphere in Dulbecco's modified Eagle's medium (DMEM) supplemented with 10% fetal bovine serum (FBS) and containing 1% penicillin/streptomycin to prevent mycoplasma contamination. Murine osteoblasts MC3T3-E1 and its derived Rr-*Bloc1s1* variant and corresponding control cells (*Bae et al., 2019*) (generously provided by the Hollien Lab) were routinely cultured at 37°C in a 5% $CO_2$ atmosphere in alpha minimum essential media with nucleosides, L-glutamine, and no ascorbic acids, supplemented with 10% FBS. Murine fibroblasts L2 cells were routinely cultured at 37°C in a 5% $CO_2$ atmosphere in F12 medium supplemented with 10% fetal calf serum

(FCS). For BMDMs, the abovementioned DMEM with 20% L929 cell supernatant, 10% FBS, and antibiotics was used. Cells were seeded in 24- or 96-well plates and cultured overnight before infection. For antibiotic protection assays, $1.25 \times 10^5$ (BMDMs) or $2.5 \times 10^5$ (RAW264.7) host cells were seeded in each well; for fluorescence microscopy assays, $1 \times 10^4$ or $5 \times 10^4$ cells were seeded in 96-well plates or on 12 mm glass coverslips (Fisherbrand) placed on the bottom of 24-well microtiter plates, respectively; for host RNA analysis, $1 \times 10^5$ host cells were seeded in each well of 24-well plates before infection. Host cells were infected with *Brucella* at an MOI of 100, unless otherwise indicated. Infected cells were then centrifuged for 5 min ($200 \times g$) and incubated at 37°C. Thirty minutes to 1 hr postinfection, culture media was removed, and the cells were rinsed with 1× phosphate buffered saline (PBS, pH 7.4). Fresh media supplemented with 50 μg/ml gentamicin was then added for 1 hr to kill extracellular bacteria. Infected cells were continuously incubated in the antibiotic. At the indicated time points post infection, viable bacteria in infected cells were analyzed using the antibiotic protection assay or the immunofluorescence microscopy assay as previously described (*Pandey et al., 2018*; *Qin et al., 2008*).

## Viral propagation, infection, and plaque assay

Wild type MHV-A59 was propagated in L2 cells in F12 media with 2% FCS. Host cells (RAW 264.7) were infected with MHV-A59 in triplicate at a MOI of 1. Infected cells were incubated at room temperature with gentle rocking for 1 hr. Afterwards, culture media was removed, and the cells were rinsed with 1× PBS (pH 7.4). Fresh media supplemented with 2% FBS was added. Infected host cells were incubated at 37°C. At the indicated time points post infection, viral supernatants were collected and then titrated by plaque assay on L2 cells at 33°C.

## Generation, genotyping, and characterization of *Ern1* CKO mice

Animal research was conducted under the auspices of approval by the Texas A&M University Institutional Animal Care and Use Committee in an Association for Assessment and Accreditation of Laboratory Animal Care International Accredited Animal Facility. To investigate the roles of IRE1α in controlling Bm16M intracellular parasitism, *Ern1* CKO (*Ern1*$^{flox/flox}$; *Lyz2*-Cre) mice were generated by crossing *Ern1*-floxed mice, in which exons 20 and 21 of *Ern1* were floxed, with *Lyz2*-Cre mice carrying the Cre recombinase inserted in the Lysozyme M (*Lyz2*) gene locus. In the resultant animals (gift from the Iwawaki lab), exons 20 and 21 of the *Ern1* gene were specifically deleted in myeloid cells, including macrophages, monocytes, and neutrophils. The *Ern1* CKO mice were genotyped using genomic DNA from tail vain to show the presence of *Cre* alleles (*Iwawaki et al., 2009*). Western blot analysis using anti-IRE1α antibodies (Novus Biologicals) and *Xbp1* splicing were performed on BMDMs from CKO and control mice to validate the absence of full-length IRE1α in CKO mice.

## BMDM harvest and cultivation

BMDMs collected from the femurs of *Ern1* CKO and control mice were cultivated in L929-cell conditioned media [DMEM medium containing 20% L929 cell supernatant, 10% (vol/vol) FBS, penicillin (100 U/ml) and streptomycin (100 U/ml)]. After 3 days of culture, nonadherent precursors were washed away, and the retained cells were propagated in fresh L929-cell conditioned media for another 4 days. BMDMs were split in 24-well plates ($2.5 \times 10^5$ cells/well) in L929-cell conditioned media and cultured at 37°C with 5% $CO_2$ overnight before use.

## Whole animal infections with *Brucella* and histologic analysis

Mice from CKO and littermate control groups (5 mice/group or treatment) were intraperitoneally infected with *B. melitensis* and *B. abortus* (Bm16M and BaS2038, respectively) with a dose of $1 \times 10^6$ CFU. At 7 and 14 dpi, infected mice were euthanized, and the bacterial burden was assessed in spleen and liver. A portion of the tissue was fixed, and paraffin embedded for histopathological examination following hematoxylin and eosin staining. Sections were evaluated and scored for lesion severity (inflammation) using the previously described scoring system (*Lacey et al., 2018*), that is, 0 = no inflammation; 1 = minimal with inflammation involving <5% of tissue; 2 = moderate with focally extensive areas of inflammation (5–25% of tissue and involving 1 or more tissues); 3 = moderate to severe with focally extensive areas of inflammation (>25% to 50% of tissue and involving multiple tissues); and 4 = severe with large confluent areas of inflammation (>50% of tissue and involving

multiple tissues). To assess Bm16M tissue burden, spleen or liver tissues were homogenized and subjected to a serial dilution. Finally, the diluted tissue homogenates (200 µl) were plated on TSA solid plates and CFUs were determined at 48- to 60-hr postincubation at 37°C in 5% $CO_2$.

## Latex bead phagocytosis assays

Phagocytosis assays for testing the phagocytic uptake and route of a substrate in the nonfunctional and RIDD-resistant Bloc1s1 variants in RAW264.7 murine macrophages were performed using the Phagocytosis Assay Kit (IgG FITC) (Cayman Chemical, USA) according to the manufacturer's instructions.

## RNA extraction and qRT-PCR analysis

RNA was extracted from host cells per instructions in the RNeasy Mini Kit (74,134 Qiagen). Complementary DNA was amplified from mRNA using the High-Capacity cDNA Reverse Transcription Kit (4368813 Applied Biosystems) per manufacturer's guidelines. For qRT-PCR of the macrophage infections of MHV strain A59, BaS19, Bm16M, or Ba2308 1/5 dilution of each cDNA was added into nuclease free water in a respective well in a 96-well plate. SYBR green (50–90 µl) with primers (5–9 µl) were put into triplicate wells of each respective primer in the same respective 96-well plate for all experiments. Primers for *Bloc1s1*, *Cd300If*, *Diras2*, *Txnip*, *miR-17-5*p, *U6,* and *Gapdh* were used (***Supplementary file 2***). The cDNA and master mix were transferred to a 384-well plate using E1 ClipTip pipettor (4672040 Thermo Scientific). The qPR-PCR was run on a CFX384 Real-Time System (BioRad).

## RNA-seq analysis

All RNA-seq reads were mapped to *Mus musculus* reference genome GRCm38.p4, release 84, which is provided by Ensembl.org, by using STAR-2.5.2a (***Dobin et al., 2013***). The aligned reads were then counted by using RSEM-1.2.29 (***Li and Dewey, 2011***). Differential Expression Analysis was performed by using DESeq-1.30.0 (***Anders and Huber, 2010***) and edgeR-3.26.6 (***McCarthy et al., 2012***; ***Robinson et al., 2010***). Differentially expressed gene (DEGs) were determined if the gene's p value (significance of differential expression) <0.05 and the absolute value of the fold change >1.5.

## Bioinformatic analysis

KEGG pathway enrichment analysis was performed by using WebGestalt-2013 (WEB-based Gene Set Analysis Toolkit) (***Wang et al., 2013***) with RIDD genes. Gene Ontology (GO) Enrichment was performed by feeding the significantly DEGs to PANTHER-v7 (***Gaudet et al., 2011***; ***Mi et al., 2010***) in all three classes: Molecular Function, Biological Process, and Cellular Component. Both KEGG pathway and GO enrichment analysis were filtered and sorted by Fisher test and Benjamini and Hochberg adjustment. The GeneCards website (***Stelzer et al., 2016***) was used to connect to the REFSEQ mRNAs from NCBI's GenBank. The FASTA option was chosen and the fasta file was saved. The fasta file was uploaded to RNAfold Web Server (Institute for Theoretical Chemistry at University of Vienna) with the fold algorithm options of minimum free energy and partition function and avoid isolated base pairs. The Forna option was then used to view the secondary structure and manually searched for a stem loop with NNNNNCNGNNGNNNNNN. Interaction network analysis of KEGG pathways and the RIDD genes (p < 0.05) identified in this work was visualized using Cytoscape (https://cytoscape.org/) as described previously (***Zhao et al., 2017***).

## Generation of nonfunctional and RIDD-resistant *Bloc1s1* variants in RAW264.7 murine macrophages

Nonfunctional *Bloc1s1* variants in RAW264.7 murine macrophages were generated using a protocol previously described (***Hoffpauir et al., 2020***). One clone containing either an amino acid deletion substitution or deletion in one of the XAT regions of murine BLOS1 (***Figure 4—figure supplement 1A***) was selected. For generation of Rr-*Bloc1s1* variant and its control (wt-*Bloc1s1*) in RAW264.7 murine macrophages, RNA was first extracted using RNeasy Plus mini-kit. *Bloc1s1* cDNA was generated from mRNA and cloned into pCR 2.1-TOPO vector. Site-directed mutagenesis was utilized to generate a g449t mutation into one of the *Bloc1s1* plasmid clones (***Figure 4—figure supplement 1B***). Both the wild-type (WT) and mutated *Bloc1s1* segments were removed from the pCR 2.1-TOPO vector and cloned into pE2n vectors. Gateway cloning was used to generate expression vectors pLenti-CMV-Hygro-DEST (w117-1)-Blos1-WT and pLenti-CMV-Hygro-DEST (w117-1)-Blos1-G449T. At every step

post cDNA creation, plasmid amplification vectors were verified via sequencing. The plasmids were then transfected into Lenti-X cells with psPAX2 and VSVG packaging plasmids. The virus was collected at 24- and 48-hr post-transfection and stored at −80°C. RAW264.7 cells were transduced with the virus and cells containing the wt- or Rr-*Bloc1s1* expression cassette were selected using hygromycin B (500 μg/ml).

## Drug treatments

Host cells were coincubated in 24-well plates with tunicamycin or 4μ8C at the indicated concentrations. Cells were treated with drugs 1 hr before, and during, infection with the indicated *Brucella* strains and incubated at 37°C with 5% $CO_2$. At the indicated time points postinfection, the treated cells were fixed with 4% formaldehyde and stained for immunofluorescence analysis or lysed to perform CFU assay or for RNA extraction assays as described above. To investigate whether the drugs inhibit *Brucella* growth, the drugs were individually added to *Brucella* TSB cultures at 37°C and incubated for 1 and 72 hr. CFU plating was used to assess bacterial growth in the presence of drugs, and thereby to evaluate the potential inhibitory effects. Host cells in which drug treatment or *Brucella* infection induced no significant differences in viability and membrane permeability as well as drugs that have no adversary effect on *Brucella* growth were used in the experiments reported in this work.

## CIM assays

Immunofluorescence microscopy staining and imaging methods to determine *Brucella* intracellular trafficking and colocalization of the bacteria and host BORC components in infected host cells were perform as previously described (*Pandey et al., 2017*; *Pandey et al., 2018*; *Qin et al., 2011*; *Qin et al., 2008*) with minor modifications. Briefly, to visualize Bm16M intracellular trafficking, the indicated host cells ($5.0 \times 10^4$ for 24-well plates and $1 \times 10^4$ for 96-well plates) were seeded on 12 mm coverslips placed on the bottom of wells of 24- or 96-well plates (without coverslip) and infected with Bm16M-GFP or Bm16M. At 0.5 (for 24-well plates) or 1 (for 96-well plates) h.p.i., the infected cells were washed with 1× PBS and fresh media containing 50 μg/ml gentamicin was added to kill extracellular bacteria. At the indicated time points post infection, the infected cells were fixed with 4% formaldehyde at 4°C for overnight or for 20 min at 37°C before confocal fluorescence or immunofluorescence microscopy analysis was performed as previously described (*Pandey et al., 2017*; *Pandey et al., 2018*; *Qin et al., 2011*; *Qin et al., 2008*). The primary antibodies used were listed in the Key Resources Table. Samples were stained with Alexa Fluor 488-conjugated, Alexa Fluor 594-conjugated, and/or Alexa Fluor 647 secondary antibody (Invitrogen/Molecular Probes, 1:1000). Acquisition of confocal images, and image processing and analyses were performed as previously described (*Pandey et al., 2017*; *Pandey et al., 2018*; *Qin et al., 2011*; *Qin et al., 2008*).

The BioTek Cytation 5 and Gen 5 software (version 3.05) were used to calculate perinuclear Lamp1 index and autophagic flux. Specifically, for perinuclear Lamp1 index, the area of the nucleus and average intensity for Lamp1 was measured, noted as $A_N$ and $Int_N$, respectively. The area of the whole cell and the average intensity for Lamp1 was measured, noted as $A_{WC}$ and $Int_{WC}$, respectively. Next the area of the cell 3 μm off from the nucleus (peripheral region) and the average intensity for Lamp1 was measured, noted as $A_C$ and $Int_C$, respectively. Then, $Int_{WCN}$ (average intensity of Lamp1 in the whole cell minus nucleus) and $Int_P$ (average intensity of Lamp1 in the perinuclear region) were calculated as the following formula:

$$Int_{WCN} = ((A_{WC} \times INT_{WC})) - (A_N \times Int_N)/(A_{WC} - A_N)$$
$$Int_P = (((A_{WC} \times INT_{WC}) - (A_N \times Int_N)) - (A_C \times Int_C))/((A_{WC} - A_N) - A_C)$$

If $Int_P/Int_{WCN} \leq 1$, which means that there is no or very little perinuclear colocalization; if $Int_P/Int_{WCN} > 1$, which means that there is perinuclear colocalization.

For autophagic activity, using BioTek Cytation 5 and Gen 5 software (version 3.05), the mean intensity of LC3b in the nucleus (M1) and in the cytoplasm (M2) were measured using a primary and secondary mask in every individual cell. In the Gen 5 software, a subpopulation analysis was carried out to identify cells that had a ratio of M2/M1 <1. From this, the perinuclear LC3b index or autophagic activity was calculated as the following formula: Perinuclear LC3b index (or autophagic activity) = (Total number of identified cells with M2/M1 <1)/Total number of the analyzed cells.

For calculation of the relative BLOS1 fluorescence integrated density (FID) per BCV, the corrected total cell fluorescence was measured by taking the integrated density (area of cells × mean fluorescence), then the number of *Brucella* in each cell and colocalization of BLOS1 and Bm16M (BCV-BLOS1$^+$, %) were counted. Cells that did not contain bacteria were removed from the calculation. The integrated density was divided by the number of bacteria in the cell to obtain the 'Total fluorescence per bacteria'. The 'relative BLOS1 FID per BCV' was then calculated by multiplying the 'Total fluorescence per bacteria' with the colocalization percentage. Since BLOS1 protein is more stable in control cells treated with 4µ8C, the value of the relative BLOS1 FID per BCV was normalized as 1.

## Protein pull-down assays and immunoblotting analysis

Pull-down assays for testing physical interaction of proteins were perform using the Pierce Co-Immunoprecipitation (Co-IP) Kit (Thermo Scientific, USA) according to the manufacturer's instructions. Preparation of protein samples and immunoblotting blot analysis were performed as described previously (*Ding et al., 2021*; *Pandey et al., 2017*; *Pandey et al., 2018*; *Qin et al., 2011*). Densitometry of blots was performed using the ImageJ (http://rsbweb.nih.gov/ij/) software package. All Westerns were performed in triplicate and representative findings are shown.

## Statistical analysis

All the quantitative data represent the mean ± standard error of mean from at least three biologically independent experiments, unless otherwise indicated. The data from controls were normalized as 1% or 100% to easily compare results from different independent experiments. The significance of the data was assessed using the Student's *t*-test (for two experimental groups), two-way analysis of variance test with Holm–Sidak's multiple comparisons, or the Kruskal–Wallis test with Dunn's multiple comparison. For the RNA-seq results, $\log_2$ fold changes were calculated and results were screened to meet the threshold ($|\log_2$FC (fold change)$| > 1$, $p < 0.05$) for selection. DEGs met the criteria and in UPRsome were included in the final lists.

## Key resources

All key resources, including bacterial strains, mammalian cell lines, reagents, etc. used in this work are listed in Key resource table.

## Acknowledgements

The authors are grateful to Drs. Takao Iwawaki (Kanazawa Medical University, Japan), Laurie H Glimcher (Cornell University, USA), R J Kaufman (Howard Hughes medical Institute at University of Michigan Medical School, USA), Julie Hollien (University of Utah, USA) for sharing *Ern1* CKO and control mice, *Xbp1* CKO and control mice, *Ern1*$^{+/+}$ and *Ern1*$^{-/-}$ MEF cells, and *Bloc1s1*– expressing and control cell lines, respectively, used in this study; to Drs. Christine McFarland, Jessica Bourquin, Todd Wisner, and Susan Gater (Texas A&M) for key support; and to Drs. Steve Fullwood and Kalika Landua (Nikon Instruments) for expert assistance with the microscopy analysis. This work is supported by the Texas A&M Clinical Science Translational Research Institute Pilot Grant CSTR2016-1, the Defense Advanced Research Projects Agency (DARPA) (HR001118A0025-FoF-FP-006), NIH (R21AI139738-01A1, 1R01AI141607-01A1, 1R21GM132705-01), the National Science Foundation (DBI 1532188, NSF0854684), the Bill Melinda Gates Foundation, and to PdF; the National Natural Science Foundation of China (# 81371773) to QMQ. The National Institute of Child Health and Human Development [RHD084339] to TAF (in part). The National Science Foundation Grant 1553281 to XQ. The content of the information does not necessarily reflect the position or the policy of the Government, and no official endorsement should be inferred.

# Additional information

## Funding

| Funder | Grant reference number | Author |
| --- | --- | --- |
| Texas A and M University | Clinical Science Translational Research Institute Pilot Grant CSTR2016-1 | Paul de Figueiredo |
| Defense Advanced Research Projects Agency | HR001118A0025-FoF-FP-006 | Paul de Figueiredo |
| National Institutes of Health | R21AI139738-01A1 | Paul de Figueiredo |
| National Institutes of Health | 1R01AI141607-01A1 | Paul de Figueiredo |
| National Institutes of Health | 1R21GM132705-01 | Paul de Figueiredo |
| National Science Foundation | DBI1532188 | Paul de Figueiredo |
| National Science Foundation | NSF0854684 | Paul de Figueiredo |
| Bill and Melinda Gates Foundation | | Paul de Figueiredo |
| National Natural Science Foundation of China | 81371773 | Qing-Ming Qin |
| National Institutes of Health | RHD084339 | Thomas A Ficht |

The funders had no role in study design, data collection, and interpretation, or the decision to submit the work for publication.

## Author contributions

Kelsey Michelle Wells, Conceptualization, Data curation, Formal analysis, Investigation, Methodology, Validation, Visualization, Writing – original draft, Writing – review and editing; Kai He, Haowu Chang, Xueqiang Li, Hao Zhang, Sing-Hoi Sze, Data curation, Formal analysis; Aseem Pandey, Ana Cabello, Dongmei Zhang, Jing Yang, Gabriel Gomez, Xuehuang Feng, Luciana Fachini da Costa, Investigation; Yue Liu, Data curation, Formal analysis, Writing – review and editing; Richard Metz, Cameron Lee Martin, Jill Skrobarczyk, Luc R Berghman, Allison Ficht, Resources; Charles D Johnson, Kristin L Patrick, Investigation, Resources; Julian Leibowitz, Methodology, Resources, Writing – review and editing; Jianxun Song, Conceptualization, Validation; Xiaoning Qian, Data curation, Formal analysis, Resources; Qing-Ming Qin, Conceptualization, Data curation, Formal analysis, Funding acquisition, Investigation, Methodology, Supervision, Validation, Visualization, Writing – original draft, Writing – review and editing; Thomas A Ficht, Conceptualization, Data curation, Formal analysis, Funding acquisition, Investigation, Methodology, Project administration, Resources, Supervision, Writing – review and editing; Paul de Figueiredo, Conceptualization, Data curation, Formal analysis, Funding acquisition, Investigation, Methodology, Project administration, Resources, Supervision, Validation, Visualization, Writing – original draft, Writing – review and editing

## Author ORCIDs

Kelsey Michelle Wells (iD) http://orcid.org/0000-0002-1657-5189
Kai He (iD) http://orcid.org/0000-0002-5805-3354
Yue Liu (iD) http://orcid.org/0000-0002-6213-5248
Hao Zhang (iD) http://orcid.org/0000-0002-2058-7123
Richard Metz (iD) http://orcid.org/0000-0002-9876-3385
Charles D Johnson (iD) http://orcid.org/0000-0003-3892-7082
Kristin L Patrick (iD) http://orcid.org/0000-0003-2442-4679
Julian Leibowitz (iD) http://orcid.org/0000-0003-3485-3506

Xiaoning Qian http://orcid.org/0000-0002-4347-2476
Qing-Ming Qin http://orcid.org/0000-0002-1723-6867
Paul de Figueiredo http://orcid.org/0000-0002-3878-5064

### Ethics

LysM-Ern1mut/mut or Xbp1-KO and control mice were generated under the auspices of approval by the Texas A & M University Institutional Animal Care and Use Committee in an Association for Assessment and Accreditation of Laboratory Animal Care International Accredited Animal Facility.

### Decision letter and Author response

Decision letter https://doi.org/10.7554/eLife.73625.sa1
Author response https://doi.org/10.7554/eLife.73625.sa2

## Additional files

### Supplementary files

- Supplementary file 1. Table S1. Recombinant plasmids constructed in this work.
- Supplementary file 2. Table S3. Primers used in this work.
- Transparent reporting form
- Source data 1. Source Western blot and agrose gel images in PDF format.
- Source data 2. Original images of Western blots and agarose gels.

### Data availability

All data generated or analyzed during this study are included in the manuscript, supporting file, and source data file.

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
