## [Editor Report]

To successfully replicate in the host cell, Brucella must evade degradation in lysosomes and traffic to the ER. This work uncovers a novel mechanism by which Brucella harnesses the host unfolded protein response to degrade Blos 1, a key regulator of lysosomal trafficking, thereby enabling pathogenic Brucella peri-nuclear/ER trafficking.

---

## [Decision Letter]

**Decision letter after peer review:**

Thank you for submitting your article "Diverse pathogens activate the host RIDD pathway to subvert BLOS1-directed immune defense" for consideration by *eLife*. Your article has been reviewed by 2 peer reviewers, and the evaluation has been overseen by a Reviewing Editor and Jos Van der Meer as the Senior Editor. The following individual involved in review of your submission has agreed to reveal their identity: Zhao-Qing Luo (Reviewer #2).

Essential revisions:

1) Additional experiments are requested based on this reviewer point: Figure 3D-G, particularly panel G. It has been established that RIDD is able to increase the stability of TXNIP mRNA by attacking the microRNA miR-17 that affects TXNIP mRNA stability (PMID: 22883233). In the absence of IRE1a nuclease activity, it is predicted that miR-17 should be stabilized, which will destabilize TXNIP mRNA. Yet, the authors saw increased TXNIP mRNA levels. It is necessary to determine how such increase occurs (by higher transcription? Is miR-17 involved here?).

2) Additional clarification/discussion/editing is needed based on the reviewer comments listed below.

*Reviewer #1 (Recommendations for the authors):*

In this work, the authors elucidate a mechanism by which the Brucella-induced UPR supports intracellular bacterial parasitism. Brucella activates IRE1-dependent RIDD, which degrades Blos1 mRNA, encoding a key regulator of lysosomal disposition within the cell. Lysosomal evasion and ER fusion is essential for Brucella replication and Blos1 degradation appears essential for both these events. At the end of the study, they note that IRE1 RNAse activity and Blos1 also seem important for murine hepatitis virus replication.

The major strengths of this work are the new insights into how Brucella manipulates a host cell process to enhance survival and adaptive trafficking within the cell. Their story is well supported through the use of pharmacologic manipulations, IRE1 RNase conditional mice, a series of Blos1 mutant cell lines (both non-functional and RNase resistant), and immunofluorescence microscopy. Overall, this work adds significantly to our understanding of Brucella intracellular parasitism.

The greatest weakness is that the story is hard to follow related to confusing wording and insufficiently tight correspondence between figure panels and text. Description of the methods is not always adequate. Section and figure titles do not always go with the subsequent text. Also, the title is misleading and emphasis on diverse pathogen utilization of this mechanism in abstract and discussion (final paragraph!) is disproportionate to the work done. Out of ~90 figure panels (not even including >60 supplemental data panels) only 2 are devoted to MHV. While the MHV data is interesting and suggestive, MHV is an extremely different pathogen with a different lifestyle. The title should reflect the work, which is almost entirely about Brucella.

More specific issues and suggestions for the authors:

Figure 1: The LPS stimulation of the bone marrow macrophages is not described in the Figure legend or in the methods. In the text you state the mice (how many?) were infected intra-peritoneally, then in the methods you state they were infected intra-nasally. Please specify which method was used. In Figure 1K, you should show standard CFU rather than the derivative "replication efficiency". The inset for 1h hpi does not add anything except tiny writing. What is the "inflammation score"?

P7. Bottom paragraph leaps around in logic: The initial sentence of the paragraph says you will examine trafficking, but then lines 21-24 go to control experiments regarding IRE1 levels and phosphorylation status that should probably have been introduced much earlier in the story. Then the next lines (25-29) are almost an exact replica of lines 24-29 on the previous page.

P8: you state that in WT, the pathogen transiently traffics through MPR+ compartments, but this is not evident on the confocal image and looks scarcely above 0 upon quantitation. Is it a timing issue?

P9: This is an example of the inadequate correlation between text and figure. You mention doing real time PCR to look at specific genes (3D-F), but the figures references in the text are "Figure 3A-C; Figure 3-Source dataset 1." It would help the reader for the figure to be laid out from top to bottom in the order it's described in the text. Also, the title of the Figure 3 "RIDD (regulated IRE1-dependent decay)-BLOS1 axis controls Brucella host cell infection" has nothing to do with Blos control of infection. This Figure is about identification of RIDD targets during infection. Control of infection does not come up until Figure 4.

There is probably more to the rationale for going after Blos than it being unknown how it regulates microbial infection.

Suppl Figure 1: There's a western blot showing compete IRE1 KO in the MEFs. This might be confusing for people who may not realize that you're talking about different IRE1 model animals. It would help in the supplemental data to show the western blot of equivalent IRE1 levels in the CKO and WT as well as intact phosphorylation following UPR induction.

On page 11, It would be helpful to let the reader know that you had characterized the cells before using them (supplemental data) – otherwise the reader hears you are going to assess effects of drug treatment and Blos on trafficking and then suddenly gets information about Cathepsin-beads and LC3. The supplement has a lot of complex data referred to in the main text. It would help the reader to know which panels go with the corresponding description. How is the "perinuclear index" determined? In Figure 4-supplement 3E LAMTOR staining is performed, but it's not described what the significance of this is until later in the text.

In all the immunofluorescent images, it is very difficult to make out the pink-red co-localization. This is especially difficult for images in Figures 5 and 6. Perhaps arrows could point out peri-nuclear vs peripheral accumulation?

Sentences that try to put too much information in one, linked with "or" phrases, are difficult to follow. For instance, p12 lines 18-22: "BCV interactions with KIF1b+ or KIF5b+, which preferentially drive lysosomes on peripheral tracks or perinuclear/ER tracks decreased or increased, respectively, in control, mBlos1, and wt-Blos1 cells; however the opposite phenomena were observed in Rr-Blos1 cells. Respectively refers to what? The following 3 conditions?

Figure 4: It would help to put the quantitation in the same order as the blot. Also, the lanes look like they're incorrectly labelled. For instance, in the wt cells, there's an increase in ac-tubulin density between untreated and 4u8C in the quantitation, but in the western, it looks like there is a lot less. Also, in adding 4u8c to Tm, shouldn't this block RIDD degradation of Blos and bring Ac-Tub back to baseline UT levels? What are the red stars vs blue stars? This should be described in the figure legend text. In 4K, shouldn't the stars above the blue bar at 24h be a different color since it's a different comparison than the pale greenish bars? What is fluorescent integrated density?

P15. Line 8 you conclude that your work explained events "supporting intracellular replication and cell-cell movement of the pathogen". I would agree with the replication, but the work does not address the latter at all.

P16. Line 20: what do you mean by "independent" perinuclear-trafficking?

Figure 7: Consider putting more labels on the graphs so we can tell which molecule was IP'd. What is the difference between SNAPINd and SNAPINm? Again, it would help to specify what the red stars refer to. In 7I, is the difference between control Blos and 24h infected Blos expression statistically significant? The 6 steps of the cartoon are not described in the legend and text only goes through 4. What is the significance of the dotted lines? What are the blue squiggles in endosomes? Why is some writing in Red?

*Reviewer #2 (Recommendations for the authors):*

By RNAi screenings, de Figueiredo and colleagues in 2008 found that IRE1α, one of the three key regulator of unfold protein response is required for intracellular replication of Brucella spp., but the mechanism underlying such requirement remains mysterious. The current study by the same group went a long way to show that regulated IRE1α-dependent decay (RIDD) mediated by the nuclease activity of IRE1a (not its kinase activity) is responsible for the phenotype. Specifically, the authors demonstrate that the nuclease activity of IRE1a induced by Brucella infection reduces the protein level of BLOS1, thus blocking the delivery of the bacterial phagosome BCV to the lysosome, and promoting its association with the endoplasmic reticulum, a process known to benefit the bacteria. Most of the conclusions were supported by strong results from multiple complementary methods (e.g. conditional KO mice, stable cell lines, etc.) with appropriate controls, and the manuscript is well written and easy to follow.

p14, line 4. From the described results, non-functional BLOS1 does not directly "promote" the trafficking of the pathogen to the ER. Maybe "permitted" or "allowed" is better.

[Editors’ note: further revisions were suggested prior to acceptance, as described below.]

Thank you for resubmitting your work entitled "*Brucella* activates the host RIDD pathway to subvert BLOS1-directed immune defense" for further consideration by *eLife*. Your revised article has been evaluated by Jos van der Meer (Senior Editor) and a Reviewing Editor.

The manuscript has been improved but there are some remaining issues that need to be addressed, as outlined below:

As indicated below, Reviewer #2 raises an important point with respect to TNXIP expression during Brucella infection. This will need to be addressed experimentally.

*Reviewer #1 (Recommendations for the authors):*

Thank you for responding so carefully to every point raised. The manuscript should be much easier to follow now and more accurate. No further requests.

*Reviewer #2 (Recommendations for the authors):*

It is surprising that the authors decided not to perform a quite straight experiment for an important conclusion of the paper. This reviewer simply asked the authors to determine whether under their experimental conditions Brucella infection induced RIDD cleaves miR-17, thus stabilizing mRNA of TXNIP (not asking for the mechanism of the potential difference in response to different ER stress cues). From the results here and published works by others, it seems that there is a mechanism to down regulate TXNIP probably in parallel to up regulation via cleaving miR-17 in cells infected with Brucella. I do not agree with the authors that this experiment is out of the scope of the current manuscript. The revised title and Abstract focus are so RIDD centered, it is necessary to put a nail on this issue by experiments. At the very least, the authors should use a previously luciferase reporter of the TXNIP promoter (PMID: 16301999) to assess how Brucella infection affects the expression. Finally, the paper by Hu et al., showed that miR-17 is NOT involved in the down regulation of TXNIP. The authors used this paper to support their conclusion but yet completely ignored it in their discussion.

---

## [Author Response]

Essential revisions:1) Additional experiments are requested based on this reviewer point: Figure 3D-G, particularly panel G. It has been established that RIDD is able to increase the stability of TXNIP mRNA by attacking the microRNA miR-17 that affects TXNIP mRNA stability (PMID: 22883233). In the absence of IRE1a nuclease activity, it is predicted that miR-17 should be stabilized, which will destabilize TXNIP mRNA. Yet, the authors saw increased TXNIP mRNA levels. It is necessary to determine how such increase occurs (by higher transcription? Is miR-17 involved here?).

The report by Lerner and colleagues shows that cells, undergoing ER stress induced by chemical compounds (thapsingargin, tunicamycin), increase TXNIP protein levels and mRNA stability by reducing levels of the TXNIP destabilizing microRNA, miR-17 (PMID: 22883233) (Lerner et al., 2012). *Brucella abortus* S2308 infection increases IRE1α nuclease activity (de Jong et al., 2013) and likely also activates RIDD (this work). Based on the Lerner et al., report (PMID: 22883233), *Brucella* infection should destabilize miR-17, resulting in stabilization of TXNIP mRNA levels. However, recently, Hu and colleagues demonstrated that in vitro cultivated host cells or mouse tissues infected by *B. abortus* S2308 dramatically reduced expression levels of TXNIP mRNA and/or protein (Hu et al., 2020). Consistent with this result, we found that *B. melitensis* 16M (Bm16M) infection increased IRE1α nuclease activity and activated RIDD, resulting in dramatic reductions in the expression levels of TXNIP mRNA. It should be noted that in the absence of IRE1α activity (m-IRE1α) or inhibition of IRE1α nuclease activity (by 4μ8C-treatment), the high expression levels of TXNIP mRNA in Bm16M infected m-IRE1α cells does not result from “increased” TXNIP mRNA levels in these cells (this expression level in fact is the basal expression level of TXNIP mRNA), but rather from the “dramatically decreased” TXNIP mRNA levels in IRE1α harboring cells (wt-IRE1α) infected by Bm16M (Revised Figure 3D-H).

The viewpoint that “RIDD is able to increase the stability of TXNIP mRNA by attacking the microRNA miR-17 that affects TXNIP mRNA stability” may be true in non-*Brucella*-infected cells (i.e., Mock and Mock + Tm)”. In the case of *Brucella* infection, however, the opposite result is observed in Hu et al., and also in our work. *Brucella* infection activates host cell RIDD activities that destabilizes both miR-17 and TXNIP. We thus infer that *Brucella* may also coordinate the synthesis and/or activities of other factors (e.g., bacterial effectors, host proteins) to control the expression of miR-17 and TXNIP, thereby blunting the programmed death of the infected cells. The outcome benefits *Brucella* intracellular parasitism since increased TXNIP mRNA or protein levels may ultimately result in programed cell death (Lerner et al., 2012). Finally, because (1) the involvement of miR-17 in *Brucella* infection has been proven by Hu and colleagues (Hu et al., 2020), and because (2) the involvement of host or pathogen factors, including miR-17, in the regulation of TXNIP mRNA/protein expression and/or related mechanisms will distract readers from our central story, we believe that additional investigation along these lines are out of scope for this work.

2) Additional clarification/discussion/editing is needed based on the reviewer comments listed below.

Additional clarification/discussion/editing has been performed in our revised manuscript to address this concern.

Reviewer #1 (Recommendations for the authors):In this work, the authors elucidate a mechanism by which the Brucella-induced UPR supports intracellular bacterial parasitism. Brucella activates IRE1-dependent RIDD, which degrades Blos1 mRNA, encoding a key regulator of lysosomal disposition within the cell. Lysosomal evasion and ER fusion is essential for Brucella replication and Blos1 degradation appears essential for both these events. At the end of the study, they note that IRE1 RNAse activity and Blos1 also seem important for murine hepatitis virus replication.The major strengths of this work are the new insights into how Brucella manipulates a host cell process to enhance survival and adaptive trafficking within the cell. Their story is well supported through the use of pharmacologic manipulations, IRE1 RNase conditional mice, a series of Blos1 mutant cell lines (both non-functional and RNase resistant), and immunofluorescence microscopy. Overall, this work adds significantly to our understanding of Brucella intracellular parasitism.

We appreciate the positive comments of the reviewer.

The greatest weakness is that the story is hard to follow related to confusing wording and insufficiently tight correspondence between figure panels and text. Description of the methods is not always adequate. Section and figure titles do not always go with the subsequent text. Also, the title is misleading and emphasis on diverse pathogen utilization of this mechanism in abstract and discussion (final paragraph!) is disproportionate to the work done. Out of ~90 figure panels (not even including >60 supplemental data panels) only 2 are devoted to MHV. While the MHV data is interesting and suggestive, MHV is an extremely different pathogen with a different lifestyle. The title should reflect the work, which is almost entirely about Brucella.

The comments are very constructive. Please note that this is an exhaustive study with a lot of data, and the suggested limit of 5,000 words presents challenges to the storytelling. That said, in our revised manuscript, we have addressed the concerns raised by the reviewers. Specifically, we have removed confusing wording, provided rationales of the interrogated hypotheses, tightened correspondence between figure panels and text, and provided more details in the methods. As per the reviewer’s suggestion, we have also focused more on the *Brucella* story in the revised manuscript and thus have rephrased the title: “*Brucella* activates the host RIDD pathway to subvert BLOS1-directed immune defense”. We believe that the revised manuscript is easier to follow.

More specific issues and suggestions for the authors:Figure 1: The LPS stimulation of the bone marrow macrophages is not described in the Figure legend or in the methods. In the text you state the mice (how many?) were infected intra-peritoneally, then in the methods you state they were infected intra-nasally. Please specify which method was used. In Figure 1K, you should show standard CFU rather than the derivative "replication efficiency". The inset for 1h hpi does not add anything except tiny writing. What is the "inflammation score"?

We appreciate the reviewer pointing out these issues. The bone marrow-derived macrophages (BMDMs) from m-IRE1α and control mice were stimulated with LPS (100 ng/ml), and the cytokine production of the treated cells was determined at 6 hr post stimulation. We have included this information in our revised Figure legends. The number of mice and the approach employed for mouse infection were corrected in our revised manuscript. In Figure 1K, the graph with the derivative “replication efficiency” has been replaced by the one with standard CFU data. The “inflammation score” has also been defined in the revised Materials and methods section.

P7. Bottom paragraph leaps around in logic: The initial sentence of the paragraph says you will examine trafficking, but then lines 21-24 go to control experiments regarding IRE1 levels and phosphorylation status that should probably have been introduced much earlier in the story. Then the next lines (25-29) are almost an exact replica of lines 24-29 on the previous page.

We appreciate the reviewer pointing out these issues. The repeated portion of the text has been rephrased in our revised manuscript. Specifically, we have rephrased “examined the intracellular trafficking and replication…” as “examined this process…”. To the best of our knowledge, both IRE1α RNAase and Kinase activities are required for *Brucella* intracellular parasitism. IRE1α RNAase activity was broadly described in the above two paragraphs. In this paragraph, we mainly described the requirement of IRE1α activity (both RNAase and Kinase) for *Brucella* intracellular parasitism. Logically, we do not see the rationale for the remark: “regarding IRE1 levels and phosphorylation status that should probably have been introduced much earlier in the story” and thus humbly request to keep the statement about IRE1α Kinase activity here.

P8: you state that in WT, the pathogen transiently traffics through MPR+ compartments, but this is not evident on the confocal image and looks scarcely above 0 upon quantitation. Is it a timing issue?

MPR^+^-BCVs are now indicated with arrows in the images. As pointed out by the reviewer, the low number of MPR^+^-BCVs may result from the late time point of the observation.

P9: This is an example of the inadequate correlation between text and figure. You mention doing real time PCR to look at specific genes (3D-F), but the figures references in the text are "Figure 3A-C; Figure 3-Source dataset 1." It would help the reader for the figure to be laid out from top to bottom in the order it's described in the text. Also, the title of the Figure 3 "RIDD (regulated IRE1-dependent decay)-BLOS1 axis controls Brucella host cell infection" has nothing to do with Blos control of infection. This Figure is about identification of RIDD targets during infection. Control of infection does not come up until Figure 4.

We very much appreciate the reviewer pointing out the incorrect figure references for the text and the improper title for Figure 3. The correct figure references should be “Figure 3D-G; Figure 3—figure supplement 3". The analysis of our candidate RIDD genes has also been rearranged in the text. We believe that these corrections render the figure panels in the order (from top to bottom) that they are described in the text. The title for Figure 3 has been rephrased as “Identification of host RIDD targets during *Brucella* infection”.

There is probably more to the rationale for going after Blos than it being unknown how it regulates microbial infection.

A more detailed rationale for selecting BLOS1 as a target for in depth investigation has been provided in our revised manuscript.

Suppl Figure 1: There's a western blot showing compete IRE1 KO in the MEFs. This might be confusing for people who may not realize that you're talking about different IRE1 model animals. It would help in the supplemental data to show the western blot of equivalent IRE1 levels in the CKO and WT as well as intact phosphorylation following UPR induction.

In fact, this supplementary figure only describes the results related to *Brucella* infection of MEFs. The protein expression and kinase phosphorylation levels of IRE1 in WT BMDMs, following UPR induction by *Brucella* infection, have been included in the Figure 1Q-T. Therefore, the related results pertaining to the response of m-IRE1α BMDMs to UPR induction (Bm16 infection) have been included in the revised Figure 1—figure supplement 2 as per the suggestion.

On page 11, It would be helpful to let the reader know that you had characterized the cells before using them (supplemental data) – otherwise the reader hears you are going to assess effects of drug treatment and Blos on trafficking and then suddenly gets information about Cathepsin-beads and LC3. The supplement has a lot of complex data referred to in the main text. It would help the reader to know which panels go with the corresponding description. How is the "perinuclear index" determined? In Figure 4-supplement 3E LAMTOR staining is performed, but it's not described what the significance of this is until later in the text.

We appreciate the suggestions by the reviewer and have addressed them in our revised manuscript. The “perinuclear index” had been defined in the Materials and methods section. “Figure 4-supplement 3E" has been replaced by one without LAMTOR staining since the result showing LAMTOR interactions with BCVs is described later in the manuscript.

In all the immunofluorescent images, it is very difficult to make out the pink-red co-localization. This is especially difficult for images in Figures 5 and 6. Perhaps arrows could point out peri-nuclear vs peripheral accumulation?

In fact, the pink-red color is the colocalization of BCVs (purple) with Lamp-1 (red) which can be distinguished in the peri-nuclear (blue) areas or in the peripheral areas of the infected cells. As per the suggestion of the reviewer, the perinuclear vs peripheral accumulation of BCVs-Lamp1 has been indicated by yellow and orange arrows, respectively, in the revised Figures.

Sentences that try to put too much information in one, linked with "or" phrases, are difficult to follow. For instance, p12 lines 18-22: "BCV interactions with KIF1b+ or KIF5b+, which preferentially drive lysosomes on peripheral tracks or perinuclear/ER tracks decreased or increased, respectively, in control, mBlos1, and wt-Blos1 cells; however the opposite phenomena were observed in Rr-Blos1 cells. Respectively refers to what? The following 3 conditions?

The long and unclear sentence has been rephrased as three simpler sentences. The “respectively” refers to how KIF1b+ driving lysosomes on peripheral tracks decreases, and KIF5b+ driving lysosomes to perinuclear/ER tracks increases.

Figure 4: It would help to put the quantitation in the same order as the blot. Also, the lanes look like they're incorrectly labelled. For instance, in the wt cells, there's an increase in ac-tubulin density between untreated and 4u8C in the quantitation, but in the western, it looks like there is a lot less. Also, in adding 4u8c to Tm, shouldn't this block RIDD degradation of Blos and bring Ac-Tub back to baseline UT levels? What are the red stars vs blue stars? This should be described in the figure legend text. In 4K, shouldn't the stars above the blue bar at 24h be a different color since it's a different comparison than the pale greenish bars? What is fluorescent integrated density?

As per the suggestion, quantification data of the blots have been put in the same order in the revised Figure 4. The lanes of blots are correctly labelled. The quantification data were from three independent experiments, not from the representative blots. Therefore, the quantification graphic bars may not exactly reflect the representative blot levels. The previous blotting images have been replaced by the more convincing ones and the quantitative bar graph has also been rearranged. Regarding the co-treatment of 4μ8C and Tm: Theoretically, co-treatment should bring Ac-Tub back to the baseline level of the untreated sample. However, in the experiments, we do not know which one (4μ8C and Tm) functions more efficiently. In both wt-*Blos1* and Rr-*Blos1* cells, co-treatments in the two cell lines showed no significant differences when compared to untreated cells. The red stars were modified to blue stars since the comparison of the different groups have been indicated by the linked lines. In panel 4K, significance of the different groups has been remarked with the link lines. The red asterisks have also been explained in the Figure legend.

P15. Line 8 you conclude that your work explained events "supporting intracellular replication and cell-cell movement of the pathogen". I would agree with the replication, but the work does not address the latter at all.

“cell-cell movement” has been removed from the statement.

P16. Line 20: what do you mean by "independent" perinuclear-trafficking?

The confusing word was removed and the meaning of "perinuclear-trafficking” has been clarified in our revised manuscript.

Figure 7: Consider putting more labels on the graphs so we can tell which molecule was IP'd. What is the difference between SNAPINd and SNAPINm? Again, it would help to specify what the red stars refer to. In 7I, is the difference between control Blos and 24h infected Blos expression statistically significant? The 6 steps of the cartoon are not described in the legend and text only goes through 4. What is the significance of the dotted lines? What are the blue squiggles in endosomes? Why is some writing in Red?

We appreciate the constructive suggestions by the reviewer. In fact, the molecules used in IP assays and the difference between SNAPIN_d_ and SNAPIN_m_ had been clearly stated in the previous figure legend. That said, to improve clarity, and as per the suggestion, more labels have been included in the revised Figure, including molecules used in the IPs and differences between SNAPIN_d_ and SNAPIN_m_. Also, the red stars have been explained in the Figure legend. Yes, the difference is statistically significant. All the 6 steps have been described in the revised text. We do not have our own data to support “the significance of the dotted lines”, as these events have been described in previous studies (Guardia et al., 2016; Pu et al., 2015; Starr et al., 2012). The blue squiggles in BCVs represent EEA1. Red ink was used to easily distinguish the presence of one of the conditions (e.g., IRE1α depletion, 4μ8C, or Rr-*Blos1*) that impairs or inhibits ER trafficking or the accumulation of BCVs.

Reviewer #2 (Recommendations for the authors):By RNAi screenings, de Figueiredo and colleagues in 2008 found that IRE1α, one of the three key regulator of unfold protein response is required for intracellular replication of Brucella spp., but the mechanism underlying such requirement remains mysterious. The current study by the same group went a long way to show that regulated IRE1α-dependent decay (RIDD) mediated by the nuclease activity of IRE1a (not its kinase activity) is responsible for the phenotype. Specifically, the authors demonstrate that the nuclease activity of IRE1a induced by Brucella infection reduces the protein level of BLOS1, thus blocking the delivery of the bacterial phagosome BCV to the lysosome, and promoting its association with the endoplasmic reticulum, a process known to benefit the bacteria. Most of the conclusions were supported by strong results from multiple complementary methods (e.g. conditional KO mice, stable cell lines, etc.) with appropriate controls, and the manuscript is well written and easy to follow.

We very much appreciate the positive comments from the reviewer.

p14, line 4. From the described results, non-functional BLOS1 does not directly "promote" the trafficking of the pathogen to the ER. Maybe "permitted" or "allowed" is better.

As per the suggestion of the reviewer, the word "promote" has been replaced by the more proper word “permitted”.

References

Bae, D., Moore, K.A., Mella, J.M., Hayashi, S.Y., and Hollien, J. (2019). Degradation of Blos1 mRNA by IRE1 repositions lysosomes and protects cells from stress. J Cell Biol 218, 1118-1127.

de Jong, M.F., Starr, T., Winter, M.G., den Hartigh, A.B., Child, R., Knodler, L.A., van Dijl, J.M., Celli, J., and Tsolis, R.M. (2013). Sensing of bacterial type IV secretion via the unfolded protein response. MBio 4, e00418-00412.

Guardia, C.M., Farias, G.G., Jia, R., Pu, J., and Bonifacino, J.S. (2016). BORC Functions Upstream of Kinesins 1 and 3 to Coordinate Regional Movement of Lysosomes along Different Microtubule Tracks. Cell Rep 17, 1950-1961.

Hu, H., Tian, M., Li, P., Guan, X., Lian, Z., Yin, Y., Shi, W., Ding, C., and Yu, S. (2020). Brucella Infection Regulates Thioredoxin-Interacting Protein Expression to Facilitate Intracellular Survival by Reducing the Production of Nitric Oxide and Reactive Oxygen Species. The Journal of Immunology 204, 632-643.

Lerner, A.G., Upton, J.-P., Praveen, P., Ghosh, R., Nakagawa, Y., Igbaria, A., Shen, S., Nguyen, V., Backes, B.J., and Heiman, M. (2012). IRE1α induces thioredoxin-interacting protein to activate the NLRP3 inflammasome and promote programmed cell death under irremediable ER stress. Cell metabolism 16, 250-264.

Pu, J., Schindler, C., Jia, R., Jarnik, M., Backlund, P., and Bonifacino, J.S. (2015). BORC, a multisubunit complex that regulates lysosome positioning. Developmental cell 33, 176-188.

Starr, T., Child, R., Wehrly, T., Hansen, B., Hwang, S., López-Otin, C., Virgin, H., and Celli, J. (2012). Selective Subversion of Autophagy Complexes Facilitates Completion of the Brucella Intracellular Cycle. Cell Host and Microbe 11, 33.

[Editors’ note: further revisions were suggested prior to acceptance, as described below.]

The manuscript has been improved but there are some remaining issues that need to be addressed, as outlined below:As indicated below, Reviewer #2 raises an important point with respect to TNXIP expression during Brucella infection. This will need to be addressed experimentally.

We performed additional experiments to determine expression of TXNIP in the wild-type (WT) and non-functional *Blos1* (m*Blos1*) RAW264.7 cell lines infected by *Brucella abortus* S2308 (BaS2308) in the absence or presence of 4μ8C, an *IRE1*α RNAse activity and RIDD inhibitor. Consist with our previous findings in bone marrow-derived macrophages (BMDMs) infected by *B. melitensis* Bm16M, we found that BaS2308 infection decreased TXNIP expression in cells harboring wild-type (WT) *Blos1*; however, block of RIDD by 4μ8C mitigated TXNIP mRNA downregulation. Moreover, TXNIP expression remained relatively unchanged even increases in the m*Blos1* cells (Author response image 1, left panel).

**Author response image 1. sa2fig1:** *Brucella* infection downregulates host *Txnip* (left panel) and miR-17 (right panel) expression. RAW264.7 macrophages harboring the wild-type (WT) or non-functional *Blos1* (m*Blos1*) were infected with *B. abortus* S2308 in the absence or presence of 4μ8C, an *IRE1*α RNAse and RIDD inhibitor, during infection. At the indicated hours post infection, the infected cells were harvested and performed real-time reverse transcription-PCR (qRT-PCR) assays to determine TXNIP mRNA and miR-17 expression. Un: uninfected cells. *, **, and ***: Significance at p < 0.05, 0.01 and 0.001, respectively. Data represent the mean ± standard error of means from at least 4 independent experiments were performed (expression levels of the indicated genes were normalized to *Gapdh* expression). Blue asterisks: comparison with those of uninfected control cells.

The result of reduction of TXNIP expression during *Brucella* infection was consistent with findings by Hu and colleagues who demonstrated that in vitro cultivated host cells or mouse tissues infected by *Ba*S2308 dramatically reduced expression levels of TXNIP mRNA and/or protein (Hu et al., 2020). Taken together, we believe that these data demonstrate that TXNIP expression is reduced during *Brucella* infection in a RIDD-dependent format.

Reviewer #1 (Recommendations for the authors):Thank you for responding so carefully to every point raised. The manuscript should be much easier to follow now and more accurate. No further requests.

The positive comments are very much appreciated.

Reviewer #2 (Recommendations for the authors):It is surprising that the authors decided not to perform a quite straight experiment for an important conclusion of the paper. This reviewer simply asked the authors to determine whether under their experimental conditions Brucella infection induced RIDD cleaves miR-17, thus stabilizing mRNA of TXNIP (not asking for the mechanism of the potential difference in response to different ER stress cues). From the results here and published works by others, it seems that there is a mechanism to down regulate TXNIP probably in parallel to up regulation via cleaving miR-17 in cells infected with Brucella. I do not agree with the authors that this experiment is out of the scope of the current manuscript. The revised title and Abstract focus are so RIDD centered, it is necessary to put a nail on this issue by experiments. At the very least, the authors should use a previously luciferase reporter of the TXNIP promoter (PMID: 16301999) to assess how Brucella infection affects the expression. Finally, the paper by Hu et al., showed that miR-17 is NOT involved in the down regulation of TXNIP. The authors used this paper to support their conclusion but yet completely ignored it in their discussion.

We appreciate the reviewer’s comments. As per the suggestion, we performed additional experiments to determine whether under our experimental conditions *Brucella*-induced RIDD cleaves miR-17, thus stabilizing mRNA of TXNIP. We elected to use qRT-PCR as a direct measure of RNA expression rather than the proposed luciferase system. Our finding demonstrates that *Brucella* infection does reduce miR-17 expression in WT *Blos1* and m*Blos1* cells (Author response image 1, right panel).

Interestingly, we found that in m*Blos1* cells, the downregulation of miR-17 expression was also induced but delayed. Our data, therefore, suggest that the downregulation of TXNIP likely occurs in fashion that is independent of miR-17 upregulation in cells infected with *Brucella*.

As per the suggestion, we have included the expression of TXNIP and miR-17 during *Brucella* infection in our revised *Results* and *Discussion section*.

References

Hu, H., Tian, M., Li, P., Guan, X., Lian, Z., Yin, Y., Shi, W., Ding, C., and Yu, S. (2020). Brucella Infection Regulates Thioredoxin-Interacting Protein Expression to Facilitate Intracellular Survival by Reducing the Production of Nitric Oxide and Reactive Oxygen Species. The Journal of Immunology 204, 632-643.